# Rotation-equivariant Graph Neural Networks for Learning Glassy Liquids Representations

Francesco Saverio Pezzicoli, Guillaume Charpiat, and François P. Landes

*Université Paris-Saclay, CNRS, INRIA, Laboratoire Interdisciplinaire*
*des Sciences du Numérique, TAU team, 91190 Gif-sur-Yvette, France*

The difficult problem of relating the static structure of glassy liquids and their dynamics is a good target for Machine Learning, an approach which excels at finding complex patterns hidden in data. Indeed, this approach is currently a hot topic in the glassy liquids community, where the state of the art consists in Graph Neural Networks (GNNs), which have great expressive power but are heavy models and lack interpretability. Inspired by recent advances in the field of Machine Learning group-equivariant representations, we build a GNN that learns a robust representation of the glass' static structure by constraining it to preserve the roto-translation (SE(3)) equivariance. We show that this constraint significantly improves the predictive power at comparable or reduced number of parameters but most importantly, improves the ability to generalize to unseen temperatures. While remaining a Deep network, our model has improved interpretability compared to other GNNs, as the action of our basic convolution layer relates directly to well-known rotation-invariant expert features. Through transfer-learning experiments displaying unprecedented performance, we demonstrate that our network learns a robust representation, which allows us to push forward the idea of a learned structural order parameter for glasses.

## INTRODUCTION

Understanding the nature of the dynamical glass transition is one of the most intriguing open problems in condensed matter. As a glass-forming liquid is cooled down towards its characteristic temperature ($T_g$), its viscosity increases by several orders of magnitude, while its structure, the spatial arrangement of the particles, does not show any obvious change. Finding the subtle changes in the structure that explain the huge variations in the dynamics (increased viscosity) seems like an obvious target for Machine Learning, which is great at finding hidden patterns in data.

Independently from this, a number of works identify patterns using expert hand-crafted features that seem quite specific to glasses where these descriptors are shown to directly correlate with the dynamics [1, 2], without any actual use of Machine Learning. Aside from Machine Learning, historically and up to the present day, physicists have inspected how some well-defined physical quantities correlate with glassy dynamics [1–7], in a manner akin to expert features (but without any learning). There are also direct pattern-recognition techniques [8–12] and a few unsupervised approaches [13–17], but the typical strategy is to learn a representation using supervised learning, *i.e.* use ML to map a given local structure to a target label that represents the local dynamics (*i.e.* a "local viscosity" measure). See the review [18] for a comparison or various machine-learned of physical quantities and how they correlate with dynamics. Since glasses are very heterogeneous when approaching the transition (a phenomenon called Dynamic Heterogeneity [19–22]), a single temperature point already displays a broad variety of labels' values, and is generally considered to be enough data to learn a good representation. We further articulate the relation between the physics of glasses and learning a representation in section V.

The quest for this good representation has prompted many works, relying on various techniques. The idea of using ML for glasses was introduced by the pioneering works of Liu *et al.* [23–26], where a rather generic set of isotropic features (describing the density around the target particle) is fed to a binary classifier (a linear Support Vector Machine (SVM)) to predict the target particle's mobility. After considering only isotropic features, they marginally improved accuracy by using angular-aware expert features, measuring bond-angles around the target particle. This simple yet robust technology alone yielded a number of physics-oriented works [27–32], which provide physical interpretations of the model's output value (named Softness). These and the other shallow learning approaches all rely on features that describe the neighborhood of a single particle to then predict this single particle's future: in all cases, there is no interaction between neighborhoods.

Independently from the glass literature, within other application fields dealing with particles living in 3D space, Deep Learning frameworks and in particular Graph Neural Networks (GNNs) were developed, with some success, quite early on (2016). Applications range from molecular properties predictions [33], to bulk materials properties prediction (MegNet) [34], or Density Functional Theory approximations [35], and most importantly for us, for glassy dynamics [36]. Indeed, designing geometrical features to capture the slight variations in the geometry of a particle's neighborhood is a daunting task, and the lesson learned from Computer Vision and the advent of Convolutional Neural Networks (CNNs) is that meta design works better than design, that is, expert features designed by hand are not as informative as the ones learned by a neural network with a suitable architecture designed by an expert. This is the point of the Deep approach, and more specifically GNNs, that are designed to build aggregated representations of the nodes'

neighborhoods. Indeed in 2020 Bapst *et al.* [36] substantially redefined the state of the art using a GNN. Although they are very effective, previous GNN approaches, that do not account for rotation-equivariance, lack interpretability due to their structure and their high number of learnable parameters.

The coarse-graining effectively performed by GNNs was then mimicked (or distilled) in an expert-features oriented approach, with the impressive result by Boattini *et al.* [37] where it was shown that a simple Ridge Regression with $\sim 1000$ rather generic, rotation-invariant features, performs equally well as the reference of that time, *i.e.* GNNs [36]). The features consist for a part of node-wise rotation-invariant descriptors, and for another, of these elementary features but averaged over neighboring nodes, over a small radius (mimicking a GNN aggregation step). There is currently some debate [15, 37–41] over whether Deep Learning is the right way to do better than such expert features, and part of the community leans on this expert side.

Here, inspired by the fast-growing field of SE(3)-equivariant networks [42–49], we build a GNN with hidden representations that are translation and rotation-equivariant (SE(3)-symmetry). Concretely, under rotation of the whole glass, the scalar properties of the particles (as mobility) remained unchanged, while the vectorial quantities (like relative positions) transform accordingly. With SE(3)-equivariant networks, the internal representations behave like such physical vectors: the representation rotates appropriately under rotation of the input. Invariant features have significantly less expressivity than equivariant ones, since they are just a subset of those. In other words, we take the best of both worlds: we combine the basic idea of symmetry-aware features, already used with some success [37], with the combinatorial and expressive power of Deep Learning. In practice, for the task of predicting the dynamical propensity of 3D Kob-Anderson mixtures, we significantly surpass the historical state of the art [36], at comparable or reduced number of parameters (depending on the task) and increased interpretability, while we perform comparably as well or better than other approaches (the details depends on the timescale considered). Importantly, the representation we learn generalizes very well across temperatures.

In the next section (sec. I) we define the task to be solved: input and output data. We then introduce all the necessary theoretical tools to build the basic SE(3)-GNN layer, explaining how they apply to our specific case (sec. II). We explain how to combine these layers into a network in sec. III. In section IV, we study the impact of various pre-processing choices on performance and we compare with other recent works. We open on interpretating the learned representation as an order parameter and experiment on the robustness of our representation in section V. We outline directions for future work in section VI. We summarize the main outcomes of this work in the conclusion, section VII.

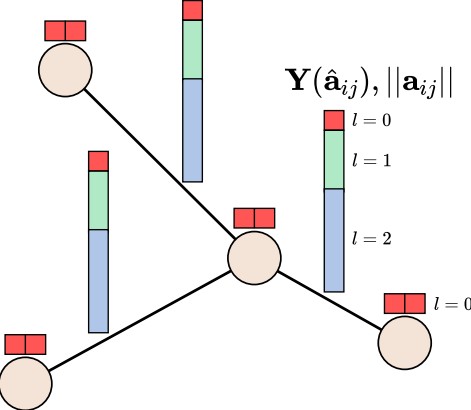

FIG. 1. **Input Graph with its input features.** Node features are the one-hot encoded particle types (invariant features, $l = 0$), and edge attributes $\mathbf{a}_{ij}$ are split: the direction is embedded in Spherical Harmonics $Y(\hat{\mathbf{a}}_{ij})$ and the norm is retained separately. Throughout this paper, we depict each rotational order with a given color: $l = 0$ (red), $l = 1$ (green), $l = 2$ (blue). The relative length of each is a reminder that each requires $2l + 1$ real values to be stored on the machine.

## I. DATASET AND TASK

To probe the ability of our model to predict mobility, we adopt the dataset built by Bapst *et al.* in [36]. It is obtained from molecular dynamics simulations of an 80:20 Kob-Andersen mixture of $N = 4096$ particles in a three-dimensional box with periodic boundary conditions, at number densities of $\rho \simeq 1.2$. Four state points (temperatures) are analyzed: $T = 0.44, 0.47, 0.50, 0.56$. For each point, 800 independent configurations $\{\mathbf{x}_i\}_{i=1...N}$ are available, *i.e.* 800 samples (each sample represents $N$ particles' positions).

The quantity to predict (Ground Truth label) is the individual mobility of each particle, measured as the dynamical propensity [50, 51]: for each initial configuration, 30 micro-canonical simulations are run independently, each with initial velocities independently sampled from the Maxwell-Boltzmann distribution. The propensity of particle $i$ over a timescale $\tau$ is then defined as the average displacement over the 30 runs. Propensity is available at $n_{times} = 10$ different timescales that span the log scale, *a priori* resulting in $n_{times}$ different tasks. For some experiments we also use another similar dataset as provided by Shiba *et al.* [38], which models the same glass-former yet differs from that of Bapst *et al.* on a couple of points, that we detail in Appendix D.

For each sample to be processed through the GNN, the input graph is built by taking particles as nodes and connecting them when the inter-atomic distance between positions $\mathbf{x}_i$ and $\mathbf{x}_j$ is less than $d_c = 2$ (in atomic potential units). The node features encode the particle type, here A or B ("Node features" is machine learning vocabulary for "set of values associated to the node", and similarly for edge features). We use one-hot encoding, such that

node features consist of $n_{type} = 2$ boolean variables. This generalizes trivially to mixtures with $n_{type} > 2$. Optionally, we also include the value of the potential energy of particle $i$ as node feature, which brings their number to 3 (2 boolean and a real). The edges are directed, and edge $(i, j)$ has for feature $\mathbf{a}_{ij} = (\mathbf{x}_j - \mathbf{x}_i)$, *i.e.* it stores the relative position of the particles (nodes) it connects. We show a sketch of our input graph with its node and edge features in Figure 1.

The task is then the node-wise regression of the particle's propensity $m_i \in \mathbb{R}$ (node label). Notably, here we simultaneously regress both particle types, meaning that all nodes contribute to the computation of the loss function. We also introduce a new task, referred to as *multi-variate regression*, in which the $n_{times}$ timescales are regressed at once, as opposed to as the usual *uni-variate* approach.

## II. HOW TO BUILD A GRAPH-CONVOLUTION EQUIVARIANT LAYER?

### A. Graph Neural Networks

Consider a graph $\mathcal{G} = (\mathcal{V}, \mathcal{E})$, where $\mathcal{V} = \{1, \ldots, n_v\}$ is the set of vertices or nodes $v_i$ and $\mathcal{E} \subseteq \mathcal{V} \times \mathcal{V}$ is the set of edges $e_{ij}$, respectively endowed with node features $\mathbf{h}_i \in \mathbb{R}^{c_v}$ and edge features $\mathbf{a}_{ij} \in \mathbb{R}^{c_e}$. GNNs operate on such graphs by updating node (and possibly edge) features through local operations on the neighborhood of each node. These operations are designed to adapt to different kinds of neighborhoods and respect node-index permutation equivariance, which are the two key features of GNNs, as opposed to CNNs (for which the learned kernels must have fixed, grid-like geometry, and for which each neighboring pixel is located at a fixed relative position). In this work we deal with Graph Convolutional Networks (GCN), a subclass of GNNs. A GCN layer acts on node features as follows:

$$\mathbf{h}'(\mathbf{x}_i) = \sum_{j \in \mathcal{N}(i)} \kappa(\mathbf{x}_j - \mathbf{x}_i)\mathbf{h}(\mathbf{x}_j) \tag{1}$$

where $\mathcal{N}(i)$ is the neighborhood of node $i$. Here a position $\mathbf{x}_i \in \mathbb{R}^3$ is associated to each node and $\kappa$ is a continuous convolution kernel which only depends on relative nodes' positions. In this case, as for CNNs, the node update operation is translation-equivariant by construction. It is however not automatically rotation-equivariant.

### B. Equivariance

A layer of a network is said to be equivariant with respect to a group $G$ if upon group action on the input, the output is transformed accordingly. One simple example is shown in Figure 2: it depicts a 2D vector field $\mathbf{f}(\mathbf{x})$ and a mapping $\mathcal{K}$ that acts on it, $\mathcal{K}(\mathbf{f}) = \cos(||\mathbf{f}||) \cdot \hat{\mathbf{f}}$ (where

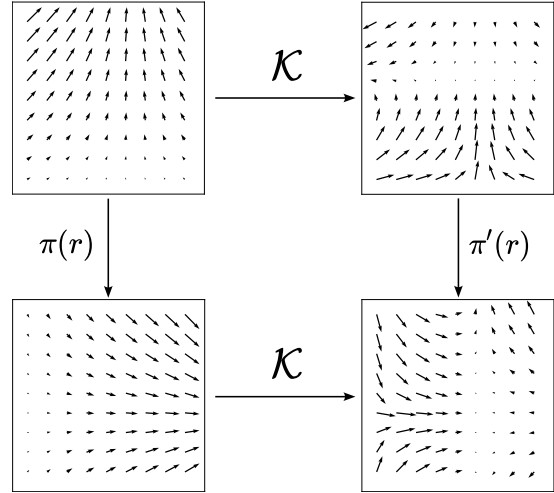

FIG. 2. **Equivariance to 2D rotations.** Simple case in which the input and output fields have the same dimension, 2. $\pi(r)$ represents the action of the rotation operator on the input field, $\pi'(r)$ on the output one. In general they can be different, here since input and output are in the same space, they are equal. The mapping $\mathcal{K}$ acts in an equivariant way, indeed it commutes with the rotation.

$\hat{\mathbf{f}} = \mathbf{f}/||\mathbf{f}||$), generating the output field $\mathbf{f}'(\mathbf{x})$, which happens to live in the same space. $\mathcal{K}$ is equivariant to 2D rotations: it operates only on the norm of the vectors, thus it commutes with the rotation operator.

As introduced in the previous example, we can represent the point cloud processed by the GNN as a vector field $\mathbf{h}(\mathbf{x}) = \sum_{i \in \mathcal{V}} \delta(\mathbf{x} - \mathbf{x}_i)\mathbf{h}_i$ with values in some vector space $H$, and the action of a layer as a mapping $\mathcal{K}$ from one field $\mathbf{h}_i$ to the updated one $\mathbf{h}'_i$. To require that $\mathcal{K}$ fulfils equivariance, we need to define how the group of interest acts on the vector space $H$ through representations. Given a group $G$, a representation $\rho$ is a mapping from group elements $g$ to square matrices $\mathbf{D}^H(g) : H \to H$ that respect the group structure. Essentially it tells how $G$ acts on a specific space $H$. For example, the representation of the group of three-dimensional rotations $SO(3)$ on the 3-D Cartesian space is the usual rotation matrix $\mathbf{R}(r_{\alpha,\beta,\gamma}) = \mathbf{R}_x(\alpha)\mathbf{R}_y(\beta)\mathbf{R}_z(\gamma)$. Then if we consider an element $tr \in SE(3)$ which is composed of a translation $t$ and a rotation $r$, it will act on a vector field as follows:

$$\mathbf{h}(\mathbf{x}) \xrightarrow{\pi(tr)} \mathbf{D}^H(r)\mathbf{h}(\mathbf{R}^{-1}(r)(\mathbf{x} - \mathbf{t})) \tag{2}$$

The codomain (output domain) is transformed by the representation of the rotation while the domain is transformed by the one of the inverse roto-translation. See Figure 2, top left to bottom left, for an example with 2D rotations. For further explanations and examples, see [44].

Let us define equivariance. Let there be a mapping $\mathcal{K} : \mathbf{h}(\mathbf{x}) \to \mathbf{h}'(\mathbf{x})$ and $\mathbf{h} \in H$, $\mathbf{h}' \in H'$ with $H, H'$ two vector spaces. The kernel $\mathcal{K}$ is equivariant with respect

to $G$ if

$$\forall g \in G \ \ \mathcal{K} \circ \pi(g) = \pi'(g) \circ \mathcal{K} \qquad (3)$$

The input and output codomains $H, H'$ do not need to be identical, and this is taken into account by the group representations $\mathbf{D}^H(g)$ and $\mathbf{D}^{H'}(g)$. A direct consequence of this definition is that invariance is a particular case of equivariance where $\mathbf{D}^{H'}(g) = \mathbb{I} \ \forall g \in G$.

When dealing with $SE(3)$, the *only* invariant quantities are *scalars*, thus considering only invariant features would significantly reduce the model's expressivity.

### C. Equivariant features

To enforce equivariance of layers, we work with equivariant features (also called *steerable features*), following the schema of steerable group convolutions [42, 44, 52] (for theoretical insight, see Appendix-B of [45]). These features inhabit the space of irreducible representations of $SO(3)$, which is factorized into sub-spaces: $V_{SO(3)} = V_{l_1} \oplus V_{l_2} \oplus V_{l_3} \oplus \dots$. Each subspace, indexed by $l \geq 0$, is of size $2l + 1$ and transforms independently under rotations thanks to the action of Wigner-D matrices $\mathbf{D}^{(l)} \in \mathbb{R}^{(2l+1) \times (2l+1)}$. Coming to implementation, a feature is just a concatenation of different $l$-vectors: scalars ($l = 0$), 3-D vectors ($l = 1$), 5-D vectors ($l = 2$) and so on. Multiple pieces with the same $l$ are also allowed, we address this multiplicity by referring to channels. For example we can have two $l = 0$ channels, a single $l = 1$ channel and a single $l = 2$ channel:

$$\mathbf{h}(\mathbf{x}) = \left( h_{c=0}^{(l=0)}(\mathbf{x}), h_{c=1}^{(l=0)}(\mathbf{x}), \mathbf{h}_{c=0}^{(l=1)}(\mathbf{x}), \mathbf{h}_{c=0}^{(l=2)}(\mathbf{x}) \right) \quad (4)$$

where $\mathbf{h} : \mathbb{R}^3 \to \mathbb{R}^k$ with $k = \sum_l n_c^{(l)} \cdot (2l+1) = 2 \cdot 1 + 3 + 5 = 10$ with $n_c^{(l)}$ number of channels of type $l$. Rotation of these features is straightforward:

$$\mathbf{D}(r)\mathbf{h} = \begin{bmatrix} D^{(0)} & & & \\ & D^{(0)} & & \\ & & \mathbf{D}^{(1)} & \\ & & & \mathbf{D}^{(2)} \end{bmatrix} \begin{bmatrix} h_{c=0}^{(l=0)} \\ h_{c=1}^{(l=0)} \\ \mathbf{h}_{c=0}^{(l=1)} \\ \mathbf{h}_{c=0}^{(l=2)} \end{bmatrix} \quad (5)$$

The representation matrix is block-diagonal thanks to $SO(3)$ being decomposed in a direct sum, and scalars ($l = 0$) being invariant with respect to rotation ($D^{(0)} = 1$). The wording *steerable* becomes clear: upon rotation of the input coordinates, these features rotate accordingly, as when one steers the steering wheel, thus turning the wheels.

### D. Spherical Harmonics (SH)

To embed the three dimensional node and edge input data in an equivariant form, we use Real Spherical Harmonics $Y_m^l : \mathbb{S}^2 \to \mathbb{R}$. They can be thought of as the generalization of Fourier modes (circular harmonics) to the sphere. Spherical Harmonics are indexed by the *rotational order* $l \geq 0$, which is reminiscent of the 1-D frequency, and by $m = -l, \dots, l$, which determines the spatial orientation. They form an orthonormal basis of $\mathbf{L}^2(\mathbb{S}^2)$, *i.e.* any real-valued function on the sphere $f : \mathbb{S}^2 \to \mathbb{R}$ can be Fourier Transformed to this SH basis:

$$\mathscr{F}(f)(l,m) = \hat{f}_m^l = \int_{\mathbb{S}_2} f(\mathbf{n}) Y_m^l(\mathbf{n}) \mathrm{d}\mathbf{n} \qquad (6)$$

$$f(\mathbf{n}) = \sum_{l=0}^{\infty} \sum_{m=-l}^{l} \hat{f}_m^l Y_m^l(\mathbf{n}) \quad \text{(inverse transform)}$$

$$(7)$$

where $\mathbf{n} = (\theta, \phi) \in \mathbb{S}$ represents a generic direction or point on the sphere. Here the coefficients $\hat{f}^l$ are not real values but are $(2l + 1)$-dimensional vectors (with components $\hat{f}_m^l$). The set of all coefficients $(\hat{f}^l)_{l=0,\dots}$ plays the same role as $\mathbf{h}(\mathbf{x})$ in Eq.(4), and each coefficient $\hat{f}^l$ transforms according to a Wigner-D matrix $\mathbf{D}^{(l)}$: the SH embedding is thus equivariant.

In particular, the density of neighbor particles at a fixed distance $r$ from the central one, $\rho_r(\mathbf{n}) = \sum_{j \in \mathcal{N}(i)} \delta(\mathbf{n} - \mathbf{n}_{ij}) \delta(r - r_{ij})$, is a real-valued function (distribution) on the sphere and can be decomposed into Spherical Harmonics. Furthermore, summing such decompositions at a number of radii $r \in [0, d_c]$, one obtains an equivariant representation of the density field around a target particle in the ball of radius $d_c$ (this is what our very first convolution performs, see below).

Note that to make a fixed-$r$ representation finite-dimensional, we need to choose a high-frequency cutoff for the rotational order, $l = l_{max}$. Analogously to Fourier transforms on the circle, this way of decomposing and filtering out high-frequencies preserves the input signal better than most naive schemes (as *e.g.* a discretization of solid angles).

### E. Clebsh-Gordan tensor product

As said above, we do not want to restrict ourselves to invariant features, but to equivariant ones. For this, we need a way to combine feature vectors together other than the dot product (which produces only invariant scalar features).

Analogously to the outer product for vectors of $\mathbb{R}^3$, which is a bilinear operator $\otimes : \mathbb{R}^3 \times \mathbb{R}^3 \to \mathbb{R}^3$, the Clebsh-Gordan tensor product $\otimes_{l_1, l_2}^{l_O} : V_{l_1} \times V_{l_2} \to V_{l_O}$ is a bilinear operator that combines two $SO(3)$ steerable features of type $l_1$ and $l_2$ and returns another steerable vector of type $l_O$. It allows to maintain equivariance when combining equivariant features: consider $\mathbf{h}_1(\mathbf{x}) \in V_{l_1}$, $\mathbf{h}_2(\mathbf{x}) \in V_{l_2}$ and their C-G tensor product $\mathbf{h}'(\mathbf{x}) = (\mathbf{h}_1(\mathbf{x}) \otimes_{l_1, l_2}^{l_O} \mathbf{h}_2(\mathbf{x})) \in V_{l_O}$; if we apply a rotation $r$, inputs will be transformed by $\mathbf{D}^{(l_1)}(r)$, $\mathbf{D}^{(l_2)}(r)$

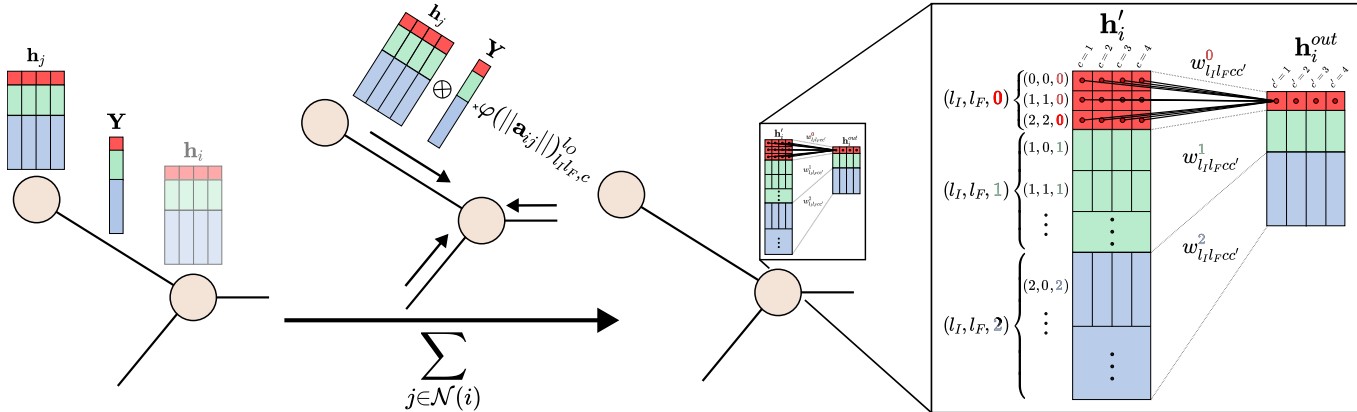

FIG. 3. **Overview of the convolution layer, summarizing Eqs. (11,12).** For each neighboring node, the node and edge features are combined (with C-G product) and multiplied by the learned radial filter $\varphi$. Before performing this operation, the one-hot encoded particle type is concatenated to $\mathbf{h_i}$ by adding 2 $l = 0$ channels (not shown, for simplicity). Because multiple triplets come out of the C-G product, we obtain a much larger representation (left part of inset). This intermediate representation is narrowed down using a linear layer (one for each $l_O$ and each channel).

and the output by $\mathbf{D}^{(l_O)}(r)$, *i.e.* equivariance is fulfilled. Concretely, the tensor product is computed using Clebsh-Gordan coefficients $C_{(l_1,m_1),(l_2,m_2)}^{(l_O,m_O)}$ as

$$h_{m_O}^{\prime l_O} = \sum_{m_1=-l_1}^{l_1} \sum_{m_2=-l_2}^{l_2} C_{(l_1,m_1),(l_2,m_2)}^{(l_O,m_O)} h_{1,m_1}^{l_1} h_{2,m_2}^{l_2} \quad (8)$$

We have $C_{(l_1,m_1),(l_2,m_2)}^{(l_O,m_O)} \neq 0$ only for $l_O \in [|l_1-l_2|, l_1+l_2]$, thus it is a sparse tensor product. In a more concise form we write:

$$\mathbf{h}^{\prime l_O} = \mathbf{h}_1^{l_1} C_{l_1 l_2}^{l_O} \mathbf{h}_2^{l_2} \quad (9)$$

where each "coefficient" $C_{l_1 l_2}^{l_O}$ is actually a $(2l_O + 1) \times (2l_1 + 1) \times (2l_2 + 1)$ tensor.

### F. SE(3)-equivariant Graph Convolution Layer

Using all the concepts introduced above, we can now define the explicit form of the convolution kernel $\kappa$ of Eq. 1. Denoting the input edge attributes $\mathbf{a}_{ij} = (\mathbf{x}_j - \mathbf{x}_i)$, the kernel factorizes the effect from its radial part $||\mathbf{a}_{ij}||$ and its directional part $\hat{\mathbf{a}}_{ij}$. Each Input rotational order $l_I$ interacts with the rotational order of the Filter $l_F$ to Output various rotational orders $l_O$. We decide to use, in a similar spirit as in [53]:

$$\kappa(\mathbf{a}_{ij}) = \varphi(||\mathbf{a}_{ij}||)_{l_I l_F,c}^{l_O} \mathbf{Y}^{l_F}(\hat{\mathbf{a}}_{ij}) \quad (10)$$

The radial filters $\varphi_{l_I l_F,c}^{l_O}$ are implemented as Multi-Layer Perceptrons (MLPs, to be learned) that share some weights among triplets $l_O, l_I, l_F$ and channels $c$ (details in Appendix A 1). Expliciting the C-G tensor product, Eq. 1 now reads:

$$\mathbf{h}_{i,c,l_I l_F}^{\prime l_O} = \sum_{j \in \mathcal{N}(i)} \varphi(||\mathbf{a}_{ij}||)_{l_I l_F,c}^{l_O} \mathbf{Y}^{l_F}(\hat{\mathbf{a}}_{ij}) C_{l_I l_F}^{l_O} \mathbf{h}_{j,c}^{l_I} \quad (11)$$

This operation is depicted in Figure 3 (left part). At this stage, operations are performed channel-wise, but $\mathbf{h}_{i,c}'$ is a concatenation of all possible triplets, and as multiple combinations of $l_I, l_F$ can contribute to a given $l_O$, it is larger than the original feature $\mathbf{h}_{i,c}$. For $l_{max} = 3$, there are 34 different triplets (instead of just 4 different values of $l$).

To go back to a reduced representation, we mix together the triplets that share the same output $l_O$, with a linear layer. However, to let the various channels interact, we also perform channel mixing (also called self-interaction) with a linear layer. As linear layers combine linearly, this can be expressed as a single linear layer (right part of Figure 3):

$$\mathbf{h}_{i,c}^{\text{out},l_O} = \sum_{l_I l_F,c'} w_{l_I l_F,cc'}^{l_O} \mathbf{h}_{i,c',l_I l_F}^{\prime l_O} \quad (12)$$

where $c'$ is the input channel's index and $c$ is the output one. Note that all operations are now performed node-wise, and independently for each $l_O$. Note that this operation fulfills equivariance because only features with the same $l_O$ are combined together, with weights that do not depend on $m$ (all elements inside a vector are multiplied by the same factor). At this point we are back to our expected node feature shape, and the convolution layer can be repeated (up to a few technical details like using Batch-Norm and adding the previous layer's representation to the newly computed one, see next section, sec. III). See Appendix A 3 for the counting of the number of weights of the MLP and of this mixing layer.

### G. Interpretation

Here we want to insist that the first layer at least has a very clear interpretation. In the first convolution, for

which the input node features consist in two $l = 0$ channels, namely the one-hot encoding of the particle type. Since $l_I = 0$, the only non-zero C-G triplets will be the ones at $l_O = l_F = 0, 1, 2, 3$. To simplify, let us first imagine that the radial filters $\varphi_{l_I=0,l_F,c}^{l_O=l_F}(||\mathbf{a_{ij}}||)$ are not learned anymore but are replaced with a Gaussian Radial Basis $B(||\mathbf{a_{ij}}||)_r$ ($r$ is the basis index), where each element of the basis (a "soft bin") is attributed to a given channel. Then, the first layer's ($L = 0$) action is to convolve the particle density $\mathbf{h}_{j,c=A,B}^{l_I=0,L=0}$ with the kernel $B(||\mathbf{a_{ij}}||)_{r=c}\mathbf{Y}^{l_F=0,1,2,3}(\hat{\mathbf{a}}_{ij})$, *i.e.* to project the density fields (one for A's and one for B's) on the direct product of the chosen Radial Basis and chosen Spherical Harmonics. This projection can be seen as an embedding. Taking the norm of this representation, one would qualitatively obtain the expert descriptors described in [37] (for an exact and more detailed derivation, see Appendix C).

In our case, the channel mixing step Eq. (12) actually comes in before taking the norm, and since it does not have to mix any triplet (since $l_O = l_F$), it only mixes channels, *i.e.* performs a linear combination of the density fields for different particle types. Furthermore, we actually learn the functions $\varphi(||\mathbf{a_{ij}}||)$, so that we linearly combine the density fields measured at different radii early on, before mixing channels or taking the norm of the representation. To conclude, the features $\mathbf{h}_{i,c}^{\text{out},l_O,L=1}$ computed by our first layer correspond to various linear combinations (1 per output channel) of the density fields at all radii $d < d_c$ and all particle types. Each SH decomposition $\mathbf{h}_{i,c}^{\text{out},L=1} = \left(\mathbf{h}_{i,c}^{\text{out},l_O,L=1}\right)_{l_O=0,\dots,l_{max}}$ corresponds to a function on the sphere, which is interpreted as the projection on a unit sphere of the densities inside the ball of radius $d_c$, weighted according to their distance from the center and particle type. In our network, we build these intermediate features $\mathbf{h}_{i,c}^{\text{out},l_O}$ but we do not rush to compute their norm (the invariant features), instead we remain at the equivariant level to combine them, keeping the computation of invariants as the last step of our network. This difference significantly improves our ability to predict mobility: see section IV for our discussion on the key elements that increase performance, or directly Appendix B for the full ablation study.

For the next layers, although the interpretation is harder to explicit as much, the spirit is the same. The representation at any layer $L$, $\mathbf{h}_{i,c}^{\text{out},l_O,L}$ can be seen as SH decompositions of functions on the sphere. The next representation $\mathbf{h}_{i,c}^{\text{out},l_O,L+1}$ is then the weighted density field of these functions. For instance, $\mathbf{h}_{i,c}^{\text{out},l_O,L=1}$ is an aggregate field of the local density fields $\mathbf{h}_{i,c}^{\text{out},l_O,L=0}$.

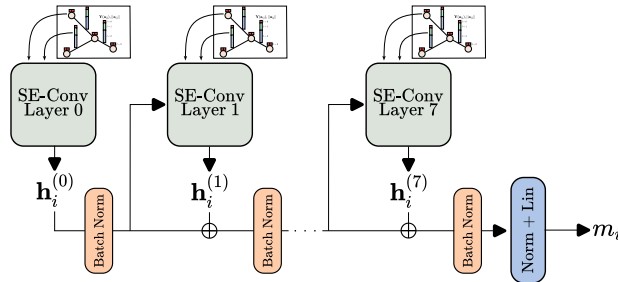

FIG. 4. **Overall Architecture.** Top: node and edge features are fed to each convolution layer. Each SE-convolution layer $L = 0, \dots, 7$ refines the output $\mathbf{h}_i^{(L)}$.

## III. COMBINING EQUIVARIANT LAYERS

### A. Network

Our network is composed of embedding blocks for nodes and edges features followed by a series of SE(3)-equivariant convolutional layers interspersed with batch normalization and connected in a Res-Net fashion [54], and one output block (decoder), as shown in Figure 4.

Here we provide a few insights on some key parts that are specific to SE(3)-equivariant networks. Further details about the architecture and the training procedure are available in Appendix A.

The code and a trained model are available on Zenodo, along with some pre-processed data, to increase reproducibility: `10.5281/zenodo.10805522`

### B. Batch Normalization

As often in Neural Networks, we sometimes need to perform Batch Normalization to avoid the neuron's activation to take overly large values. However, using a usual batch normalization layer [55] separately on each entry of the hidden representations $\mathbf{h}$ would kill the equivariance property. Thus a modified version is implemented and applied to node features [44, 56]. The $l = 0$ features are invariant and can be processed as usual:

$$h_{BN}^0 = \frac{h^0 - \bar{h}}{\sigma}\beta + \gamma \tag{13}$$

where $\bar{h} = \langle h^0 \rangle$ and $\sigma^2 = \langle {h^0}^2 \rangle - \langle h^0 \rangle^2$ with $\langle \cdot \rangle$ batch average computed with 0.5 momentum (keeping memory of previous batches) and $\beta, \gamma$ are learned parameters. For each piece of feature with $l \geq 0$, only the norm can be modified:

$$\mathbf{h}_{BN}^l = \mathbf{h}^l \frac{||\mathbf{h}^l||}{\sigma^l}\beta^l \tag{14}$$

where $\sigma^l = \sqrt{\langle ||\mathbf{h}^l||^2 \rangle}/\sqrt{2l+1}$ and $\beta^l$ are learnable parameters. In Figure 4 we show where this Batch Norm is used.

## C. Decoder

After the last convolution layer, the ultimate output block performs two node-wise operations to decode the last layer's output into a mobility prediction. First it computes SE(3)-invariant features from the hidden representation $\mathbf{h}^{(L_{max})}$: for each channel $c = 1, \ldots, 8$, the norm of each directional ($l \geq 1$) feature is computed: $||\mathbf{h}^l||_2 = \sqrt{\sum_m h_m^{l}{}^2}$, and all these norms are concatenated together with the $l = 0$ features (already invariant). Thus, we obtain an invariant representation of exactly $l_{max} + 1$ ($l$ values) $\times 8$ (channels) $= 32$ (components), which we denote $|\mathbf{h}^{(L_{max})}|$ for simplicity, despite the fact that the $l = 0$ components can be negative. The second operation is to feed this representation into a decoder, which we chose to be a linear layer, it outputs one real value, which is the predicted mobility for a given timescale and given particle type. For instance, at the timescale $\tau_\alpha$ and for particles of type $A$, the model writes:

$$y_{A,\tau_\alpha} = \mathbf{w}_{A,\tau_\alpha} |\mathbf{h}^{(L_{max})}(\{\mathbf{x}\})|, \qquad (15)$$

where $y_{A,\tau_\alpha}$ is the mobility label and $\mathbf{w}_{A,\tau_\alpha}$ is a set of weights to be regressed (32 real values). In the multi-variate setup we regress mobilities at all timescales at once, using one linear decoder (set of weights $\mathbf{w}$) per timescale and per particle type (20 different decoders for the Bapst dataset).

## D. Non-linearities

We note that all the layers act linearly on the node features. The only non-linearities of the network are hidden in the implementation of the radial part of the filters $\varphi$ (MLPs). This limited scope of non-linearities is unusual, and is needed to preserve equivariance (as pointed out above when we describe Batch Norm). We have explored other forms of non linearities, like Gate-activation, without observing significant improvement.

## IV. EXPERIMENTS, RESULTS

Here we report on the performance of our architecture, discuss the role of the task, input choices and architecture choices, and compare with recent works that tackle the same problem.

## A. Experimental setup

To increase our model's robustness, we simultaneously predict the mobility both for A and B particles, instead of focusing only on the A's. The accuracy turns out to be similar for the two types. Here we show results only for one type, A, which is the reference most other works are also using. As in the works we compare with, we use the Pearson correlation coefficient as performance metric, which is invariant under shift and scale of the test labels distribution. The network architecture and hyper-parameter choices were optimized for a single task ($T = 0.44$ and $\tau = \tau_\alpha$ for uni-variate and $T = 0.44$ for multi-variate), using only the train and validation sets. The resulting choices were applied straightforwardly to other tasks, thus preventing over-tuning of the hyperparameters. The number of convolution layers is 8, thus the last representation is indexed $L = L_{max} = 7$ (representation $\mathbf{h}_i^{(L=0)}$ at $L = 0$ is the input, before any convolution). At each layer $L > 0$ the internal or hidden representation $\mathbf{h}_i^{(L)}$ has a maximum rotational order $l_{max} = 3$ and a number $n_c^{(l)} = n_c^{(0)} = n_c^{(1)} = n_c^{(2)} = n_c^{(3)}$ of channels, $n_c = 4$ for uni-variate and $n_c = 8$ for multi-variate. These choices arise from striking a balance between over- and under-fitting, under our compute-time and memory budget constraints.

Note that we perform a different train-test split with respect to [36], which does not explicitly use a test set. Here, for each state point, 400 configurations are used for training, 320 for validation and 80 for the final test.

In Appendix A, we provide more details about the training of the model.

## B. Uni-variate or Multi-variate

In Figure 5 we compare the performances of various choices for our model, in particular uni-variate and multi-variate approach (red triangle and red diamonds). We see that we get almost the same prediction accuracy by training only one model instead of ten models, provided we increase the number of parameters for that single model: we double the number of channels in the multi-variate case, from 4 to 8, thus going from $\sim 25000$ to $\sim 50000$ parameters, see appendix A 2 for the precise counting. In other setups we even see slightly increased performance when comparing multi-variate multi-particle regression with uni-variate, A particles only regression. In any case, we observe that the multi-variate choice slightly improves the robustness of our representation: it generalizes better to other temperatures. Beyond performance considerations, it is very advantageous when considering generalization to other temperatures, since all timescales are encompassed in the same representation $|\mathbf{h}^{(L_{max})}|$. In this sense, our network is about an order of magnitude less parameter-hungry that other models, where each of the 10 timescales and each particle type need a dedicated network.

## C. Role of Inherent Structures

It has been observed several times that pre-processing the input positions by quenching them to their corre-

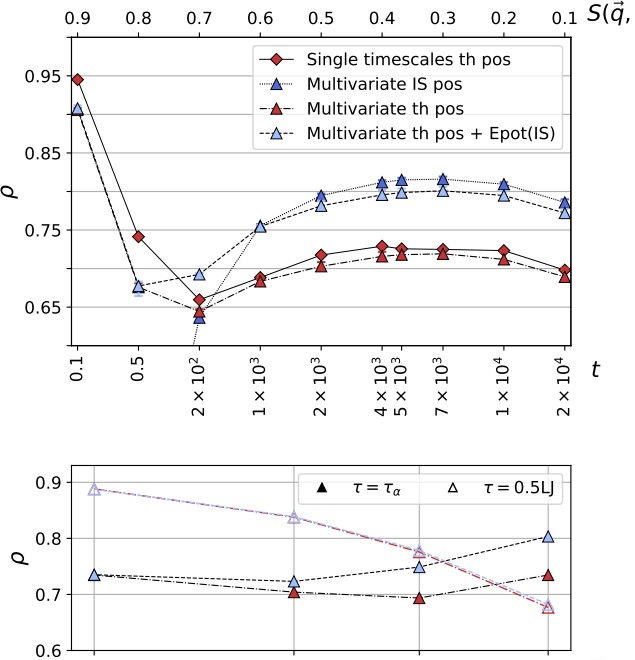

FIG. 5. **Multi-variate vs uni-variate and influence of inputs.** (top) Correlation $\rho$ between the true and the predicted propensity for A particles at temperature $T = 0.44$ as function of timescale. Marker shapes distinguish multivariate and uni-variate approaches. Colors picture the input type: red for thermal positions ($\{\mathbf{x}_i^{th}\}$), blue for quenched (Inherent Structures, IS) positions ($\{\mathbf{x}_i^{IS}\}$) and light-blue for combined ($\{\mathbf{x}_i^{th}\} + E_{pot}^{IS}$). Error-bars represent the best and the worst $\rho$ for ten identical models trained independently with different random seed initialisation, and are comparable with marker's sizes. (bottom) Correlation $\rho$ as function of training (and testing) temperature. Two timescales are shown: $\tau = \tau_\alpha$ (full markers) and $\tau = 0.5\tau_{LJ}$ (empty markers). Color code and marker code identical to that of the top plot. The multi-variate, thermal positions $+ E_{pot}(IS)$ choice is a good compromise to maintain high performance across timescales.

sponding Inherent Structures (IS) helps most Machine Learning models in predicting long-time mobility measures [26, 39, 40]. Such a quench is performed using the FIRE algorithm: temperature is set to 0 (velocities set to 0), and positions adjust gradually so as to converge to a local minimum of the potential energy, typically close to the original configuration. This can be seen as a mere pre-processing step (for which the knowledge of the interaction potentials is needed) or as a new task, *i.e.* predicting the propensities $\{m_i\}$ from the quenched positions $\{\mathbf{x}_i^{IS}\}$. We note that the quench, while intuitively cleaning some noise related to thermal motion, destroys information too: one cannot recover the thermal positions from the quenched one (the mapping thermal to quenched is non injective).

We observe that for our network, this new task is harder at short timescales, while it's easier at long

timescales (in Figure 5, compare the red diamonds and the dark blue downward-pointing triangles). We interpret this result by noting that the quench destroyed the information about the relative location of each particle within its cage, thus making it much harder to predict short-time displacements. Our experiment and its interpretation explain why some models, based on quenched positions alone, have very low performance at short timescales [39]. Their low performance should not be attributed to the machine learning models themselves, but rather to their input data. About mobility at long times, there it is not much of a surprise that quenched positions reveal an underlying slowly-evolving pattern in the structure and thus help at prediction (although in principle all the information was contained in the original thermal positions).

Ideally, one would like to combine both the complete information from thermal positions and the de-noised information from the quenched positions. For GNNs, this could be done by building the graph from either the thermal or quenched relative positions, but using as edge features a concatenation of both. However this would be quite costly in terms of memory and would increase the number of parameters needlessly. Instead, inspired by the findings of [39], we compute the local potential energy (in either the thermal or the IS positions) for each particle $E_{pot,i} = \sum_{j \neq i} V_{LJ}(\mathbf{x}_i, \mathbf{x}_j)$ and use it as a new scalar ($l = 0$) input node feature. This can be seen as a compressed version of the positional information. Note that the first layer remains very interpretable: this new channel represents the field of potential energies surrounding a given particle, expressed in the spherical harmonics basis. In Table I we compare performances obtained for all combinations of input positions (thermal or quenched), with all possible $E_{pot}$ inputs (none, thermal or quenched), resulting in 6 combinations, that we study at two timescales: $0.5\tau_{LJ}$ and $\tau_\alpha$. We summarize the key results from this table:

- Adding the information about $E_{pot}^{IS}$ to $\{\mathbf{x}_i^{IS}\}$ is irrelevant. Indeed we observed that we could easily regress $E_{pot}^{IS}$ from a network with $\{\mathbf{x}_i^{IS}\}$ input with very high precision ($\rho \approx 0.9$).

- Similarly for thermal positions and thermal potential: adding $E_{pot}^{th}$ to $\{\mathbf{x}_i^{th}\}$ is basically useless, the

| | $t = 0.5LJ$ | | $t = 1\tau_\alpha$ | |
|---|---|---|---|---|
| | $\{\mathbf{x}_i^{th}\}$ | $\{\mathbf{x}_i^{IS}\}$ | $\{\mathbf{x}_i^{th}\}$ | $\{\mathbf{x}_i^{IS}\}$ |
| no $E_{pot}$ | $0.676^{+0.005}_{-0.006}$ | $0.274^{+0.001}_{-0.002}$ | $0.718^{+0.007}_{-0.003}$ | $0.815^{+0.003}_{-0.003}$ |
| $E_{pot}^{th}$ | $0.678^{+0.006}_{-0.014}$ | $0.334^{+0.002}_{-0.001}$ | $0.728^{+0.005}_{-0.003}$ | $0.816^{+0.003}_{-0.001}$ |
| $E_{pot}^{IS}$ | $0.677^{+0.005}_{-0.013}$ | $0.273^{+0.001}_{-0.002}$ | $0.798^{+0.005}_{-0.002}$ | $0.822^{+0.003}_{-0.001}$ |

TABLE I. **Influence of IS at low temperature.** For each combination of inputs, a multi-time model is trained at temperature $T = 0.44$. We repeat the training 10 times with variable parameter initialization and report the test set correlation coefficient (median, best and worst values).

increase from $\rho = 0.718$ to $0.728$ is barely statistically significant.

- Adding $E_{pot}^{th}$ to $\{\mathbf{x}_i^{IS}\}$ helps only at short timescales (from $\rho = 0.27$ to $0.33$) and it's not sufficient to fill the gap with thermal positions.

- Adding $E_{pot}^{IS}$ to $\{\mathbf{x}_i^{th}\}$ helps, but at long timescales only (from $\rho = 0.72$ to $0.80$)

- For predicting short times, thermal positions work much better than quenched ones: 1st column shows consistently larger performance than the 2nd one, by up to 0.4 more in correlation.

- For predicting long times, quenched positions work better than thermal ones: 4th column shows consistently larger performance than the 3rd one, by up to 0.1 more in correlation.

- A good compromise for maintaining performance at all timescales is to combine $E_{pot}^{IS}$ to $\{\mathbf{x}_i^{th}\}$.

In the table we focus on two timescales for clarity, and in Figure 5 (top) we report results for 3 out of the 6 combinations but at all times. In Figure 5 (bottom) we study the effect of adding $E_{pot}^{IS}$ to the thermal positions (red to blue symbols) as a function of temperature, for two timescales. We verify that for the long timescale (full symbols) the addition of $E_{pot}^{IS}$ helps especially for the lower temperatures, where the potential energy landscape is expected to be more relevant, while for the short timescale (open symbols) there's no improvement at all, at any temperature.

We can compare these observations with the findings of Alkemade *et al.* [40]. They identify three physical quantities, each being relevant in a given time range:

1. In the ballistic regime, the forces $\mathbf{F}_i = -\nabla_{\mathbf{x}_i} E_{pot,i}$ are most relevant

2. In the early caging time, the distance between the thermal position and the IS one $\Delta r^{IS}$ is most relevant

3. At later times, the quenched configurations are most relevant

For the ballistic regime, our results perfectly match theirs: our model is likely to be aware of information equivalent to the forces, since it's able to regress the local potential energy with very high accuracy ($\rho \approx 0.9$). This explains our good performances in the very early regime (see also Figure 6). For the early caging regime, we tried to introduce $\Delta r^{IS}$ as a further $l = 0$ node feature but were not able to see any significant improvement in the caging regime. This may be due to improper encoding of this information, or to a deeper shortcoming of our architecture, or also to the datasets being slightly different (see Appendix D). For the long times, our performances are indeed high thanks to the use of $E_{pot}^{IS}$: they are slightly higher if we use $\{\mathbf{x}_i^{IS}\}$ (see Table I or Figure 5 (top)).

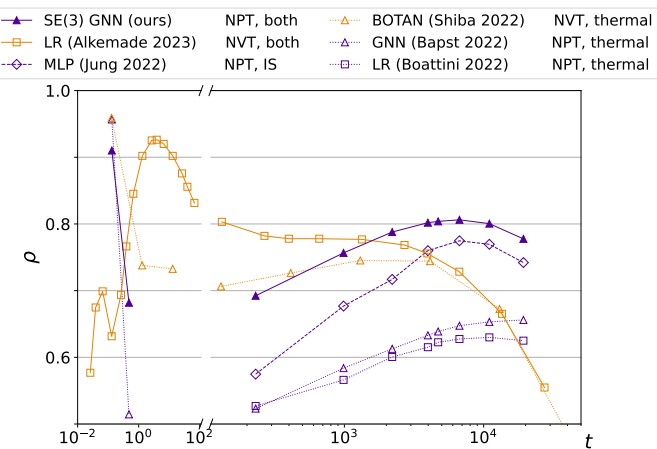

FIG. 6. **Comparison with recent works.** Correlation $\rho$ between true and predicted propensity for A particles at temperature $T = 0.44$, as a function of timescale, for several recent works. Color indicates dataset choice: dark purple for Bapst' (NPT equilibration), orange for simulations using NVT equilibration. Line-styles indicate input choices: thermal data only (dotted lines), IS data (dashed lines) or a combination of both (solid line). Markers describe the type of model: upper triangles refer to GNNs, diamonds for MLPs and squares for Linear Regression. The grey shaded area locates $\tau_\alpha$ which is slightly different for the two datasets. Note how curves computed for a given dataset barely cross each other, indicating rather consistent ranking between models.

### D. Comparison with recent works

Often, comparison with previous works can not be done rigorously, for two reasons: use of different datasets and different input data. As mentioned in the previous section and already pointed out in [40] the two main datasets [36, 38] differ in many features (see Appendix D for details), although being built from MD simulations of the same system (3D Kob-Andersen mixture). A detailed comparison at fixed input dataset is presented in a Roadmap paper [57] by other authors and some of us. A further difference is introduced by the choice of input data. For instance we have shown that the introduction of Inherent Structures helps, especially for low temperatures and long timescales. Thus better performances for works that rely on IS do not directly imply that the machine learning architecture is better (or vice versa for works that are limited to thermal inputs).

Despite these limitations, in Figure 6 we provide a qualitative comparison of methods by considering each as a whole, regardless of the details of dataset and input choice. Thus we compare our model trained on thermal positions + $E_{pot}(IS)$ at temperature $T = 0.44$, with recent works in the field [36–40].
Boattini *et al.* [37] and Alkemade *et al.* [40] apply linear regression techniques on expert rotation-invariant features: structural descriptors that capture the local density and express it in terms of spherical harmonics.

They also perform coarsening of these descriptors by averaging over the neighborhood of each particle, in a manner reminiscent of the aggregation step of a GNN. In particular [40] emphasizes the significance of IS inputs for long timescale predictions and that of using the estimated cage size as input for short timescales.

Jung *et al.* [39] apply an MLP on coarse-grained structural indicators computed from quenched positions including local potential energy and estimated dynamical fluctuations and introduce a carefully designed loss function to capture the spatial correlation of the original mobility field.

Bapst *et al.* [36] and Shiba *et al.* [38] apply Graph Neural Networks (GNN) on raw inputs, as we do here. The latter study introduces an auxiliary task of edge-regression: to enhance the accuracy of mobility predictions, they regress the change in particles' relative positions.

Our proposed approach outperforms all previous methods for timescales approaching the structural relaxation time ($\tau_\alpha$), while demonstrating competitive results in other regimes. Notably, our model achieves comparable performance to other GNN approaches on short timescales (ballistic motion), despite being the first to regress all timescales simultaneously. For the early caging regime, we do not perform as well as Alkemade *et al.* [40] although it is important to note that they incorporate early-times related information as an input feature. To be fair, we overperform Shiba *et al.* [38] only when using the quenched input.

Jiang *et al.* [41] (not shown) use a GNN that computes angles between nearest-neighbors (*i.e.* it provides geometric information, reminiscent of our equivariant approach), and introduce a self-attention mechanism that is designed to capture the spatial heterogeneity of the dynamics, referred to as smoothness in their work. It is not clear to us whether their network is partly equivariant or not, and it is rather obviously heavier than ours. At most timescales we perform a bit better.

Since the first version of this paper (preprint of Nov. 2022), we tried to include these recent works' ideas to improve performance. Our use of $E_{pot}$, inspired by [39], was indeed successful. However, when we tried to mimick [38] by regressing the edge relative elongation as an additional (edge) target label, or when we tried to reproduce the results of [40], using as input node feature the distance to the local cage center (estimated as the quenched position), or when we introduced equivariant attention schemes (inspired by [41] but technically as in [58, 59] or [60]), our attempts did not yield any significant improvement (nor deteriorated the performance).

To compare Machine Learning architectures in a fair way, one should work at fixed task (fixed dataset and input data). We now respect this constraint to obtain two precise results, that we deem significant.

Firstly, going back to using only the thermal positions as input, we perform an ablation study on the choice of $l_{max}$, to compared fairly with Bapst *et al.*  , and no-

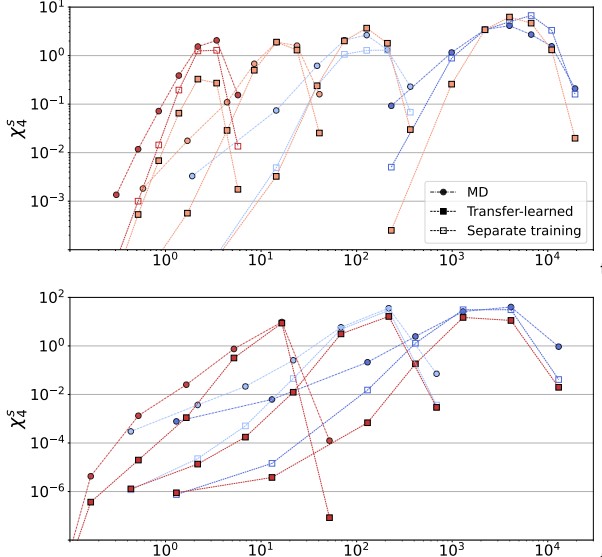

FIG. 7. **Fluctuations of the Self-overlap function.** Time evolution of the fluctuations as measured by $\chi_4^s(t)$. Top: On Bapst's dataset; Bottom: Shiba's dataset. MD is short for Molecular Dynamics and refers to the ground truth. "Separate training" indicates a new model was trained at each temperature (but dealt with all timescales at once), while "transfer-learned" refers to sec. V: we apply a single model trained at a given temperature to all other temperatures (top: $T_{train} = 0.50$, bottom: $T_{train} = 0.56$).

tice that: (i) restricted to $l_{max} = 0$, we reach the same accuracy, (ii) increasing $l_{max}$ notably improves results, especially up to $l_{max} = 2$. We conclude that the equivariant nature of a network can be key to its performance, compared to previous GNN approaches. Numerical proof is provided in Appendix B, figure 11.

Secondly, using the same kind of (invariant) inputs as non-GNN methods [37, 39, 40], *i.e.* thermal positions combined with $E_{pot}(IS)$, we study the impact of the network's depth. We noticed already in Figure 6 that we perform better than those methods, at most timescales. Here we want to stress that the network's depth plays a crucial role (more so that the rotational order $l_{max}$): varying the number of convolution layers from $L_{max} = 1$ to $L_{max} = 7$, we noticed that performance does not even saturate. We conclude that although using invariant features (and ideally, equivariant ones) is helpful, the combinatorial power of deep learning architectures is also key to performance. Numerical proof is provided in Appendix B, figures 12 and 13.

A side result of these ablation studies is that the short timescales seem to be the ones that benefit the most from increased $l_{max}$, while they also benefit from increased depth ($L_{max}$). We conjecture that directional features are key to computing instantaneous forces, itself a key element for predicting short-time dynamics.

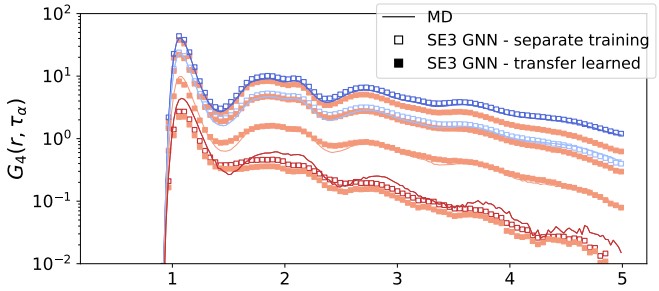

FIG. 8. **Spatial dynamical correlations.** The function $G_4$ is computed on the true labels (MD, solid line) or on our predictions. Same color and marker coding as previous plot. Our models reproduce $G_4$ remarkably well ("separate training"), especially at low temperatures (blue), while the transfer-learned fields track the trends and orders of magnitude correctly as well.

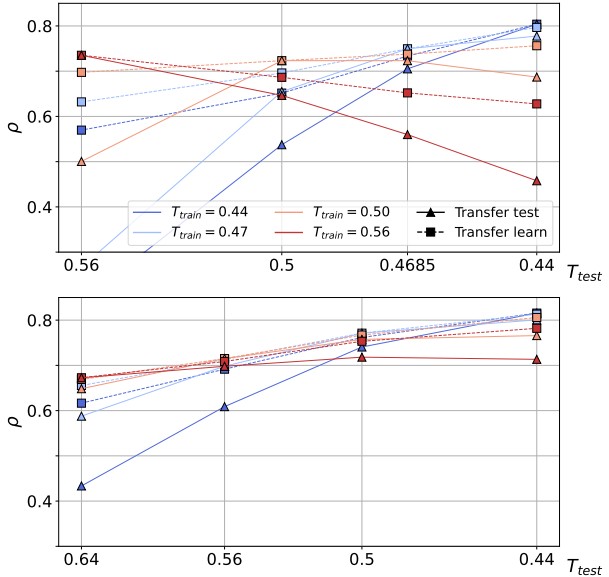

FIG. 9. **Transfer-learning between different temperatures.** Each model is fully trained once at one state point (T) and tested ("transfer test") or fine-tuned on the remaining ones ("transfer learn") . The timescale of mobilities showed in the plot is $\tau = \tau_\alpha(T)$, but multi-times models were used. (top) Bapst' dataset, (bottom) Shiba's dataset. For each training temperature (color) two different experiments are performed: transfer test (square markers with dashed lines) and transfer learn (upper triangle with full line). This results in 8 curves per plot coming from all the combinations of colors and line-styles. The transfer learned-generalization on Shiba's dataset is almost indistinguishable from direct training, indicating excellent generalization power of our learnt representation.

## E. Spatio-temporal correlations

The particle-wise correlation coefficient between ground truth mobility and predicted one is not everything, it's good to also measure whether the statistical properties of our predicted mobility match those of the true one. Defining $c_i(t) = \tanh\left(20(m_i(t) - 0.44) + 1\right)/2$ a pseudo-binarized mobility measure, $Q^s(t) = \frac{1}{N_A}\sum_{i\in A} c_i(t)$ its sample average (also called self-overlap function), one defines a four-point correlation function $\chi_4^s(t) = N_A\left[\left\langle Q^s(t)^2\right\rangle - \left\langle Q^s(t)\right\rangle^2\right]$, the fluctuations of the Self-overlap function, that we report in Figure 7 (we use the same specifications as in [39]). This measure of the sample-to-sample fluctuations of mobility is often interpreted as a volume of correlation (as it can be re-written as the integral of a correlation function). Our estimated $\chi_4^s$ ("separate training") is generally smaller than the ground-truth (MD) but tracks variations over time fairly well and much better so than the initial GNN of [36]. Furthermore it is comparable to the performance of [39], which however incorporates information about fluctuations in theirs model's loss. One may notice that the amplitude of fluctuations is smaller in the first dataset (Baspt's): this is due to the peculiar sampling choice, in which samples at a given "timescale" are actually taken at different times but equal value of the self-intermediate scattering function $F_k(t)$, a choice which by definition reduces the variance between samples.

A complementary measure of the statistical quality of the predicted mobility field is given by the spatial correlation function of the mobility-related quantity $c_i(t)$:
$$G_4(\mathbf{r}, t) = \frac{V}{N_A}\left\langle\sum_{i,j\in A} \tilde{c}_i(t)\tilde{c}_j(t)\delta(\mathbf{r} - \mathbf{r}_i(0) + \mathbf{r}_j(0))\right\rangle$$
where $\tilde{c}_i(t) = c_i(t) - \langle c(t)\rangle$. Our predictions reproduce it almost perfectly (see Figure 8).

## V. TEMPERATURE GENERALIZATION

Here we want to push forward the idea that one may be able to use deep learning to define a structural order parameter. We start by developing further the arguments briefly evoked in the introduction.

Although it is not clear yet whether the glass transition is indeed a thermodynamic phase transition or a crossover, an avoided transition, there are signs of a dynamical phase transition [12]. Under cooling through the critical temperature, a sample goes from a fully liquid to an almost fully solid state (provided the measurement timescale for defining liquid/solid is set constant) and around the transition temperature, there is a coexistence of mobile (liquid-like) parts and non mobile (solid-like) parts, reminiscent of phase coexistence. Swap-Monte Carlo methods have allowed to measure this effect while remaining in equilibrium, which confirms that this scenario is not an effect of falling out of equilibrium [61]. This simultaneous presence of active and passive regions is referred to as Dynamic Heterogeneities, which are dynamic because they relate to the local dynamical state and also because this dynamical state slowly fluctuates

over time. The deep learning program for glasses is then quite clear: define a machine learning model, *i.e.* a function $f_\theta(\{\mathbf{x}\})$ that solely depends on the structure as input, and train it to predict the local mobility (such as propensity, which acts as a proxy for the dynamical order parameter, which has a sharp crossover from active to passive). Thus one obtains a formally purely structure-based function $f_\theta(\{\mathbf{x}\})$ that has a rapid change around $T_d$, *i.e.* is reminiscent of a structural order parameter. A counter-argument to this line of thought is that by definition such an order parameter is not strictly structure-based, because it uses mobility as training data and since "neural networks can overfit", this function $f_\theta(\{\mathbf{x}\})$ simply tracks what it was trained to fit, *i.e.* mobility: in a sense, regressing mobility would be like "cheating". Indeed, it is conceivable that a heavy network with millions of parameters, that would be specialized to a given temperature and timescale, could associate mobility variations to tiny peculiarities of the physics of that particular temperature and timescale, and it would not generalize to other temperatures or timescales. The underlying idea is also that if the function $f_\theta$ is so heavy and complicated that it cannot be interpreted, we do not learn anything about the physics, or at least it is not a satisfying order parameter. In this view, a network that would reach a correlation of $\rho = 1$ would be seen as a simple computational shortcut to compute the iso-configurational displacement faster than Molecular Dynamics (which in itself would already be quite useful, *e.g.* for designing effective glass models, as [32]).

However, here we argue that a Deep Learning model $f_\theta$ should be seen as a microscope that magnifies the subtle structural variations present in the structure. Training a model to predict mobility is just a recipe to extract the relevant structural variations, but the details of this training are irrelevant. To reconcile our view with the previous one, self-supervised learning is a promising route. We recall that an order parameter $f(\{\mathbf{x}\})$ must be defined uniquely for a given system, regardless of temperatures, which translates into applying the same (trained) model $f_\theta$ to all temperatures, at least for a given glass-former. Then, a test to check whether this function does track the relevant structural changes is to measure its ability to predict mobility and its spatial and temporal correlations, especially at temperatures other than the training one. If proven, this ability for temperature generalization would contradict the main point of the argument that "the neural network overfits". This kind of simple transfer-test performance measure was already introduced a few years ago [30, 36], while the idea of applying the trained model to other temperatures dates back from the original works on Machine Learning applied to glasses [23].

## A. Transfer-testing

Here we repeat this experiment and observe better temperature-generalization abilities of our network, as compared with the original work of Bapst [36], as shown in Figure 9 (top part, label "Transfer test"). We also perform the same experiment using the more recent Shiba's dataset [38], showing even better temperature-generalization. This is a strong indication that our network learns the relevant subtle structural signatures rather than "overfitting" the dynamics separately at each temperature. We note that performance at a given temperature decreases as the training temperature goes further away from the test temperature (reading markers vertically). This can be attributed either to an increasing dissimilarity in the structures present, or also to a change in how these structures correlate with the dynamics at different temperatures. We also note an asymmetry in the performance drop between training at high temperature, testing at low (red line) or vice versa (blue line): the training at high temperature generalizes better, comparatively. In works based on SVM, the opposite was observed, and attributed to the noisy nature of high-temperature data. Here we do not seem to suffer from this noise, and attribute the increased generalizability to the larger diversity in input structures observed and the broader range of propensities observed when training on high temperature data.

## B. Transfer-learning

Here we go further, embracing the Deep Learning notion of learning *representations*, and comment on the properties of our learned representation. Indeed, the convolution layers that we stack, as described in section III, effectively build an equivariant feature $\mathbf{h}^{(L_{max})}$ that describes the local structure around each particle. As explained in sec. III (*decoder*), the norm $|\mathbf{h}^{(L_{max})}|$ of these features is a list of 32 numbers (8 channels times 4 possible $l$ values, $l = 0, 1, 2, 3$) that is decoded into mobility by 20 independent decoders. Thus, for any training temperature the model has to somehow pack the information about these 20 (non-independent) scalar values into the 32 components of $|\mathbf{h}^{(L_{max})}|$.

From such a representation, one can consider doing various things. For instance, one can perform clustering to study the number of different local structures, or perform dimensional reduction (*e.g.* PCA) to visualize their distribution. A cluster of points $|\mathbf{h}_i^{(L_{max})}|$ in the 32-dimensional space could then be seen as a Machine-Learned Locally Favored Structure (LFS). It is then physically informative to track the evolution of the population of each cluster [8, 62] (type of local structure) for instance as a function of temperature or time since a quench. Although here we use labels (mobility) to learn our representation, it is in principle possible to adopt a self-supervised learning strategy to learn a representation

$|\mathbf{h}^{(L_{max})}|$ only from input data (spatial configurations), using a pretext task as *e.g.* de-noising particles' positions. Note that the original works of Liu *et al.* were also focusing on a representation rather than a model's prediction, when using the continuous output of the SVM as a physical quantity of interest (named Softness), instead of using its thresholded, binarized version (which is the actual prediction of an SVM classifier), although in that case it is a one-dimensional representation.

Here we further probe the robustness of our model by testing the generalization ability of its underlying representation, $|\mathbf{h}^{(L_{max})}|$. If we consider $|\mathbf{h}^{(L_{max})}|$ to be a generic structural descriptor, a sort of multi-dimensional structural order parameter, then it must be relevant at all temperatures. A simple way to evaluate whether this structural measure captures the glass transition is to see whether it tracks dynamics correctly, especially in temperatures different from the training one. Concretely, we train a representation $|\mathbf{h}^{(L_{max})}|$ by regressing labels at a given temperature, and then fine-tune only the decoders at other temperatures. The part of the network responsible for computing $|\mathbf{h}^{(L_{max})}|$ (most of it) is frozen, so the fine-tuning reduces exactly to linear regressions (we need to learn the weights $\mathbf{w}$ of the decoders as in Eq. 15, *i.e.* only 32 values per timescale and per particle type). This idea of transfer learning is central to Machine Learning, and has shown great success *e.g.* in computer vision. For instance, a Convolutional Neural Network (CNN) is trained on a first task (*e.g.* ImageNet data, with 1000 classes). Then the backbone of the network (all convolution layers) is frozen, and the last 2 layers that decode this representation into labels are re-trained, but for another task (*e.g.* CIFAR10, or any other kind of natural images). This transfer-learning experiment can show improved performance compared to directly training the network on the final task, especially when there are less data available there [63]. Most importantly, the fact that transfer learning can perform well is an indication that the representation learnt by the network is more generic than one could think: the backbone is good at extracting image's features, it can be thought of as an advanced image pre-processing tool. An application of transfer learning is few-shots learning [64]: having build a good representation from a first large dataset (either with labels or with self-supervised learning), one then trains a classifier using only a handful (1 to 5) examples (per class). In our case, $|\mathbf{h}^{(L_{max})}|$ is good at extracting structural features, or more precisely at detecting patterns that correlate with the mobility.

We report the results of our own transfer-learning experiment between temperatures in Figure 9. As expected, the performance (dashed lines) is improved compared to transfer-testing (full lines). Since the predictions are computed as a linear combination of the components of $|\mathbf{h}^{(L_{max})}|$, the correlation coefficient $\rho(y_{A,\tau_\alpha}, y_{A,\tau_\alpha}^{GT})$ shown here can be seen as an aggregated measure of the correlations between the ground-truth labels $y_{A,\tau_\alpha}^{GT}$ and the individual components of $|\mathbf{h}^{(L_{max})}|$.

Going further into the direction of considering $|\mathbf{h}^{(L_{max})}|$ as a (multi-dimensional) equivalent of a structural order parameter, one could then study how the coefficients $\mathbf{w}_{A,\tau_\alpha}$ depend on the target temperature, and *e.g.* attempt to fit them with interpretable functions of the temperature. We note that among the components of $\mathbf{w}_{A,\tau_\alpha}$, most of them vary monotonously with temperature, and in particular all the $l = 0$ components do (those dominate the total). We leave deeper study of these coefficients for future works.

As a further check of our representation's robustness, we also report the transfer-learned estimated $\chi_4^s$'s in Figure 7, which show larger discrepancies than those trained at each temperature, but still track the trends seen in the data (similarly in Figure 8). Note that this transfer-learned $\chi_4^s$ measures the structural heterogeneity, since our input is purely structural, using a single set of descriptors (the representation $|\mathbf{h}^{(L_{max})}|$). To our knowledge, this is the first time a unique set of descriptor is shown to display such large structural fluctuations across temperatures and times. Here we do not show the transfer-test results for clarity (they are typically a bit worse).

## VI.  FUTURE DIRECTIONS

Here we performed non-exhaustive but rather thorough architecture search, and found no performance gain when trying rather obvious improvements for the network, such as increasing $l_{max}$, increasing the decoders' complexity, introducing bottleneck layers (reduced $l_{max}$ or channel number), using attention mechanisms as in [47, 58, 59], or attributing channels to specific bond-types. This list of negative results does not prevent us from formulating further suggestions for improved performance, which can apply to our model or other's, and are left for future work. This includes:

- regressing not the iso-configurational average displacement, but single trajectories-specific displacements (in norm). From a computational perspective, using an average (the propensity) to train, when 30 individual realizations are available, appears as throwing away some of the training data. With the mean squared error loss, training on single instances will converge to a model that is good at predicting the average.

- fully using the equivariant nature of the network to predict the vector of the displacement (3 components in 3D) instead of a scalar (its norm); this needs to be combined to the first idea.

- performing a sort of data augmentation by adding noise to the input. In practice, a very good "noise" would be to sample the positions at short timescales around $t = 0$, *e.g.* at $t = 0.5\tau_{LJ}, 1\tau_{LJ}, 1.5\tau_{LJ}, 2\tau_{LJ}, \ldots$. For predicting

timescales $\tau \gg \tau_{LJ}$, this is a negligible perturbation and would allow to teach the network what are similar configurations.

- decode various timescales with a single decoder that would be timescale-aware, in a fashion akin to that introduced in FiLM [65] for instance (conditioning the decoder to a proper embedding of the timescale as in [66], so as to use a single final decoder).

- training the backbone on several temperatures at once, using separate decoders for separate temperatures (possibly using the previous idea also for decoding temperatures, so as to have a single decoder, that would be timescale-aware and temperature-aware).

- in the spirit of [39], use non-local quantities as additional target labels (additional terms in the loss), such as the global value of correlation functions evaluated at a few lengths (label computed for the whole sample, resulting in a graph-wide target), or more simply the local variance of the mobility (variance of target label for a node's neighborhood). This is expected to increase the quality of the prediction in terms of spatio-temporal correlations, *i.e.* decrease over-smoothing, a known issue in GNNs.

- use coarse-grained mobility measures as target labels. Indeed, they have been shown to be more structure-dependent [50] and thus also display larger correlation coefficients with simple structural descriptors [67]. Eliminating noise in the labels could help achieve better precision, possibly reaching the maximum.

- use more expressive equivariant architectures, such as those recently introduced in [46].

Self-supervised learning is a possible way around the ontological issue that our structural features are trained using dynamical data (as labels). Here we outline a few possible self-supervised strategies:

- Contrastive learning: a network could be made to identify when 2 configurations are almost the same (input configurations differing by a couple of $\tau_{LJ}$) as opposed to independent configurations.

- Denoising: adding nonphysical noise to thermal (or quenched) positions, ask the network to denoise the input.

- Predict only known quantities such as $E_{pot}$, or the quenched/thermal positions, from the input thermal/quenched positions.

A much more ambitious (and debatable) idea would be to use a very heavy backbone (with attention for instance) and mix tasks between various glass-forming liquids, various crystalline materials, together with other

amorphous material's input, to require the backbone to generalize even more strongly. This kind of pre-training strategy has been shown [68–70] to be effective to improve robustness.

Whatever improvements one could think of, we believe that the SE(3)-GNN framework is the right one to be developed. Indeed, it seems in line with the recent history of neural architecture design [71]: while the CNN architecture has been a game-changer thanks to its enforcement of translation-equivariance by design, the GNNs then enforced the node permutation-equivariance by construction, and SE(3)-GNNs now additionally enforce the rotation-equivariance, leveraging all the symmetries at hand.

## VII. CONCLUSION

In this paper we first provide a pedagogical introduction to the general theoretical framework of rotation-equivariant Graph Neural Networks, then present an original Neural Network design specifically tailored to model glassy liquids. In particular, inspired by recent works on Machine Learning glasses [39, 40], we combine information from thermal positions and their quenched counterpart, using the local potential energy of quenched positions as input, thus boosting our network's performance. We disentangle the role of thermal and quenched positions: the former are necessary to predict dynamics at short times, the latter are helpful at long times. The potential energy itself is useless to us, it can be predicted accurately from positions alone, we only use it as a shortcut to pass information about the quenched positions.

As is well known in physics, finding symmetries and enforcing them in a model is key to reducing the number of degrees of freedom. In the machine learning vocabulary this translates to building representations that respect the proper symmetries [71]. SE(3)-equivariant networks achieve this by combining two ingredients. First, the proper embedding by Spherical Harmonics builds a description of the neighborhood that is both easy-to-rotate and compact since it is a decomposition in rotational order, akin to frequency decomposition, a much more efficient representation than directly discretizing angles. Second, layers combine the hidden representations $\mathbf{h}_i^{(L)}$ in a way that preserves equivariance up to the last layer, a choice that guarantees maximum expressivity for the network as opposed to rushing to invariant features. These two ingredients are key to building a good representation: our overall performance is above that of other approaches at most timescales and we achieve better generalization across tasks, while using fewer learnable parameters.

More precisely, we compare well with two families of architectures. On the one hand, compared to Deep Learning and in particular GNN models *i.e.* models that are not equivariant, nor even invariant [36, 38], our SE(3)-equivariant architecture performs better with much fewer parameters, as soon as we use strictly equivariant fea-

tures ($l_{max} > 0$), *i.e.* we prove the usefulness of equivariance. On the other hand, compared to shallow learning techniques [37, 39, 40] that use expert features as input (invariant features and the local potential), our deep network performs better (when enriched with the combined information from thermal and quenched positions), and the deeper it is the better it performs, *i.e.* we prove the usefulness of deep architectures (as embodied by GNNs).

In terms of interpretability, while we cannot claim our network to be fully interpretable, we show that our first hidden feature $\mathbf{h}_i^{(L=1)}$ corresponds to representing the field of density locally, around node $i$, *i.e.* it relates directly to the Bond Order parameter (BO) variant introduced in [37]). The next layer is a field of that local density field representation, *i.e.* much less intuitive to grasp, yet much easier to describe with explicit formula than the representations typically built by usual GNNs (which rely on fully connected layers to compute representations, thus completely entangling the inputs).

Last but not least, in this paper we emphasize the importance of building a robust representation: as explained in sec. V, the pure performance measured by the correlation of our predictions with the ground truth mobility is a means to an end, not an end in itself. What truly matters is for our representation of the local structure to allow to deduce physical facts. Our good correlation $\rho$, the very good fit of $G_4$ and acceptable trends of predicted $\chi_4$'s are all clues that we built a decent representation: we are able to capture the mobility field locally as well as its spatial and temporal correlations. But most crucially, the fact that a representation learnt at a given temperature can readily generalize to other temperatures is what allows one to consider this representation as more than a learned structural descriptor, but something akin to an acceptable structural order parameter. This generalization power is due mostly to our use of an equivariant representation, and is reinforced by our idea of regressing all particle types and all timescales at once: we use a single backbone representation, the various predictions differing only in the final decoder. Furthermore, inspired by recent success in machine learning, we introduce a new way to think about the network's output: rather than fo-

cusing on the scalar prediction, we discuss the role of the representation $|\mathbf{h}^{(L_{max})}|$. We present an example use of $|\mathbf{h}^{(L_{max})}|$: it can be correlated linearly to the target mobility, performing almost equally well as a fully retrained network for one dataset, thus showing the generalization power of this representation. Further physical study of this representation is left for future work.

The present work focuses on building a representation for glassy materials, but we would like to stress that progress in this area is intimately connected to progress made in other application areas, whenever the input data consists in particles living in 3D space (as in *ab initio* or effective potential approximations, crystalline materials or amorphous materials' properties prediction), regardless of the precise task or output label. While each of these application tasks may need fine-tuning of the architecture to perform well, we believe that they are essentially different facets of the same problem, that is, efficient learning of representations for sets of particles in space.

The code and a trained model are available on Zenodo, along with some pre-processed data, to increase reproducibility: 10.5281/zenodo.10805522

## ACKNOWLEDGMENTS

We are thankful to [36] for sharing their dataset publicly, a good practice that should be fostered in the physics' community.
We are thankful to the e3nn library developpers [56] for developing their E(3)-GNN library, which should give a huge boost to work in this area.
We thank Erik Bekkers for his notes on Group Equivariant Deep Learning https://uvagedl.github.io/
We are thankful to the pyTorch Geometric developpers [72] (GNN library). This work is supported by a public grant overseen by the French National Research Agency (ANR) through the program UDOPIA, project funded by the ANR-20-THIA-0013-01. This work was granted access to the HPC resources of IDRIS under the allocation 2022-AD011014066 made by GENCI.

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

## Appendix A: Training details, Training curve and time/energy cost

Here we provide more details about our architecture, hyper-parameters and how the model is trained.

### 1. Radial MLP

As mentioned in the main text, the radial MLPs are the only part of the network with non-linearities. They implement the radial dependence of convolution filters $\varphi^{l_O}_{l_I l_F, c}(||\mathbf{a}_{ij}||)$, thus they take as input the norms of relative node positions. Before being fed to the MLP, each norm $||a_{ij}||$ is expanded from a real value to an array through an embedding. Here we use a Bessel basis embedding:

$$B_n(r) = \sqrt{\frac{2}{r_c}} \frac{\sin\left(n\pi \frac{r}{r_c}\right)}{r} \tag{A1}$$

where $n$ is the number of roots of each basis function: $n = 1, 2, \ldots, N_b$ and we use $N_b = 10$, $r_c = d_c = 2$. Other embeddings could be for instance a Gaussian basis (with cutoff), which would act as a kind of smooth one-hot encoding of the value $r$. In practice, the Bessel basis (which is orthogonal) has better generalization properties.

The embedded input is processed through an MLP with layers sizes $(N_b, 16, n_{comb})$ and ReLu non linearities. The output size $n_{comb}$ is the number of possible triplets (combinations), times the number of channels. We also use BatchNorm (BN) [55] and Dropout (with rate $p = 0.3$) in this MLP to stabilize training and reduce overfitting. In summary, for each combination of triplets and channels $(l_O, l_I, l_F, c)$, we have a real output

$$\varphi^{l_O}_{l_I l_F, c} = \sigma(W_{n_{comb}, 16} Dropout(BN(\sigma(W_{16, N_b} B(||\mathbf{a}_{ij}||))))) \tag{A2}$$

where the $W$'s are weight matrices and $\sigma(z) = max(0, z)$. There are also bias parameters, which are not displayed here. Note that up to the layer of 16 neurons, the MLP is the same for all triplets and channels, only the last linear layer introduces different weights for each combination.

### 2. Number of parameters

This counting refers to the version of our network with 8 channels and no $E_{pot}$ in input. In total, the MLPs of our network (across all layers) account for a number of 35 664 learnable parameters: in each layer $L > 0$ we have one radial MLP of size $(10, 16, 284)$ with 5036 parameters, for the layer $L = 0$ the MLP is of size $(10, 16, 3 \times 4)$ with 412 parameters. The other main source of learnable parameters in the Network is the part of mixing the channels (right part of fig. 3), which accounts for 16000 learnable parameters: 2272 for each $L > 0$ layer and $12 \times 8 = 96$ for the $L = 0$ layer. The total number of parameters to build the representation is thus $35664 + 16000 = 51664$. One has to add to this number the parameters of the 20 decoders (10 timescales for A and B particles). The final number is then: $51664 + 32 \times 20 = 52304$. When single variate regression is performed (as in the other GNNs works we compare with), the number of channels is reduced to 4 and the total number of parameters amounts to 23210.

### 3. Overall Architecture

We do not repeat here what is written in the main text, section III Note that our Res-Net style of update is possible when two consecutive layers have the same number of channels and same $l_{max}$. Our architecture choice is found empirically to be the stable at train time. Our architecture is built and trained in the framework of PyTorch Geometric [72] which handles all the generic graph operations. All the SE(3)-related operations (SH embedding, C-G tensor product, equivariant batch-normalization) are integrated in this framework thanks to the *e3nn* library [56].

### 4. Training strategy

In Figure 10 we display one learning curve (as function of iterations, epochs). Each epoch is a sweep over the entire 400 samples dataset (each sample represents $N = 4096$ atoms).

For training, we use the Adam optimizer with initial learning rate $\gamma = 10^{-3}$, moments $\beta_1 = 0.99, \beta_2 = 0.999$ and weight decay $\lambda = 10^{-7}$. We also add a learning rate scheduler that divides $\gamma$ by 2 at several epochs as shown by the vertical dashed lines in Figure 10. Most of the results shown in the main text are obtained with a number of epochs $n_{epochs} = 100$, this choice results from

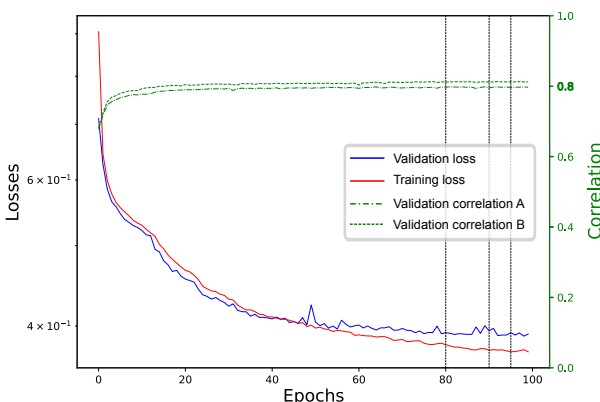

FIG. 10. **Loss and $\rho$ vs epoch.** Training of multi-time model performed at $T = 0.44$. The loss curves (full lines) correspond to the total loss of the multi-variate regression setting: sum over all the timescales of the mean squared error per timescale. The correlation curves show the Pearson correlation coefficient between predicted mobility and the ground-truth for a single timescale $\tau = \tau_\alpha$. Vertical dashed lines locate the epochs at which the learning rate is divided by 2.

several tests and strikes the balance between accuracy and training time. As it can be seen in Figure 10, each training stops before any serious overfit kicks in.

Each training of our model takes approximately 10 hours on a A100 Nvidia GPU. This represents approximately 2 kWh per training, and in France, an equivalent of 150 gCO2 (we use a number of 75 gCO2/kWh, after considering quickly EDF's optimistic figure of 25 g/kWh or RTE's more detailed daily numbers, which oscillate mostly between 50 and 100, depending on the season and time of day). We did not include the footprint of manufacturing the GPU and other infrastructure, which is generally estimated to be one half of the IT-related CO2 emissions (the other half being running the infrastructure).

## Appendix B: Ablation studies

Here we display the ablation studies, that outline which are the key elements of our model. We also report the learning curve (ablation on training set size).

### 1. Ablation of $l_{max}$.

All our results rely on the embedding of the input data into the Spherical Harmonics basis and on the built-in equivariance of convolution layers. One may expect that a large cutoff rotational order $l_{max}$ is needed. Here we show that actually, going from $l_{max} = 0$ to $l_{max} = 1$ is the most critical step. We build architectures that vary only by their $l_{max}$ value and measure the performance

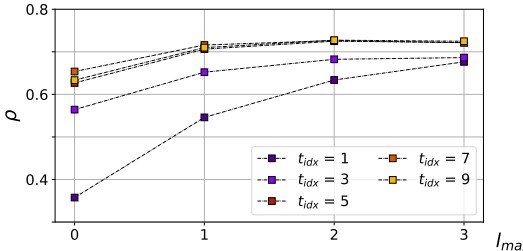

FIG. 11. $l_{max}$ **ablation using thermal positions and no $E_{pot}$ input.** A separate model was trained at $T = 0.44$, for each value of $l_{max}$. The colors correspond to the $n_{times} = 10$ timescales of mobility, with $t_{idx}$ ranging from 0 to 9. For clarity, only some of these timescales are displayed here. The GNN of [36] obtains $\rho \approx 0.65$ for the timescale 6, we are in this range when using only invariant features ($l_{max} = 0$). When using higher orders, we outperform it.

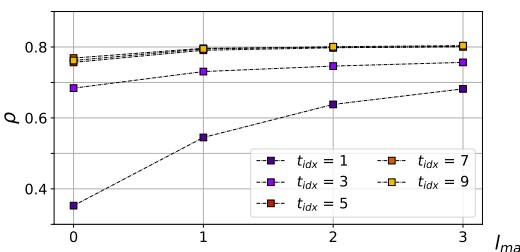

FIG. 12. $l_{max}$ **ablation** Same setup as previous plot, but here the model is using thermal positions combined with $E_{pot}(IS)$. Performance is overall higher, the relative gain from using $l > 0$ is less pronounced, probably because $E_{pot}(IS)$ already provides equivalent information.

$\rho$ in each, as shown in Figure 11 and 12. The biggest gap in performance is indeed observed between purely isotropic, scalar features ($l_{max} = 0$) versus directional ones ($l_{max} = 1$). We notice as well that short timescales require higher rotational order and the performance indeed has not saturated for them. One possible interpretation is that the network has to learn interatomic forces to describe the dynamics at short times, and that directional information is more relevant in that case. Further increasing $l_{max}$ provides a finer rotational order resolution, but we observe that the accuracy tends to saturate. We cannot go above $l_{max} = 3$ due to memory constraints: as the rotational order increases, the number of activations of neurons to keep track of grows exponentially with $l_{max}$ (while it grows linearly with the number of edges, with the batch size, and with the size of the hidden layer in the radial MLP).

### 2. Ablation of $L_{max}$ (depth).

In Figure 13 we present the performance of our multi-time model trained at temperature $T = 0.44$ for an increasing number $L_{max}$ of equivariant convolution layers

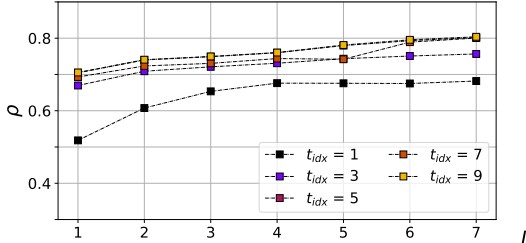

FIG. 13. $L_{max}$ **ablation.** Color-code is the same as for the other ablation studies, but here we vary the number of convolution layers applied. $L_{max} = 1$ corresponds to a very interpretable model, similar to using a Bond Order parameter and linear regression (hidden in $\varphi$ and in the decoder). Performance does not seem to saturate: one expects increased performance with more layers.

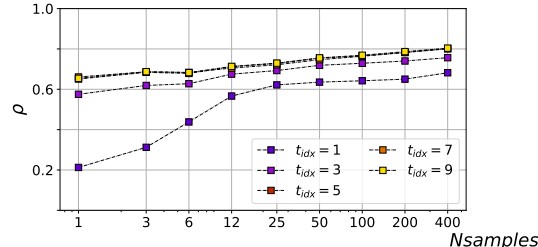

FIG. 14. **Learning curve** Color-code is the same as for the other ablation studies, but here we vary the number of training samples. Performance is already very high when training on a single sample: our network seems to resist well to overfitting. Here again, performance does not seem to saturate, more precisely it seems to increase logarithmically with train set size.

stacked in the architecture. While the short timescales seem to be saturated, the longer ones seem not: indeed, we'd expect increased accuracy if we increased $L_{max}$ further. Note that there is a possibility of encountering over-smoothing effects with an increased number of layers.

### 3. Ablation of $N_{train}$ (Learning curve).

In Figure 14, we present learning curves that illustrate the model's performance (multi-variate setting) as a function of the number of samples in the training set for different timescales. The choice of the samples to include in the training set is performed in an incremental way: for each point of the curve new samples are added to the others already present, while the test set (80 samples) is kept constant. In this version we used early stopping with a validation set of 320 samples, however using the last epoch's model yields very similar result. In [57] we used the last epoch's model and observe similar behavior. We emphasize that competitive performances are achieved already by using less than $1/4$ of the available training set and meaningful prediction are obtained also when training the model on a single sample, contrary to what one would expect for a "deep" model like ours.

## Appendix C: Reproducing expert features (Boattini 2021)

Here we relate the SE(3)-equivariant formalism with the expert features such as those used in [37]. We start by considering embedded node features $h_{i,c} = \delta_{t_i,c}$ where $t_i$ is the type of particle $i$: we have only two channels at $l = 0$. We extend them from $n_{ch} = 2 \rightarrow n_{ch} = 2 * N_b$ just by replicating them $N_b$ times. We will denote the replicated $h_{i,c}$ as $h_{i,c,r}$ since in spirit, each copy of the two channels (one-hot of the particle type) will correspond to one radial basis function. Then the convolution operation

reads (remember that for $l_I = 0$, only $l_O = l_F$ is allowed):

$$\mathbf{h}_{i,c,r,l_I=0,l_F}^{'l_F} = \sum_{j \in \mathcal{N}(i)} \varphi(||\mathbf{a}_{ij}||)_{l_I=0,l_F,c,r}^{l_F} \mathbf{Y}^{l_F}(\hat{\mathbf{a}}_{ij}) h_{j,c,r} \tag{C1}$$

We may choose a Gaussian radial basis function $B$ (instead of Bessel): $\varphi(||\mathbf{a}_{ij}||)_{l_I=0l_F,c,r}^{l_F} = B(||\mathbf{a_{ij}}||)_r$. Then if we focus on $l_F = 0$, since $Y^0(\hat{\mathbf{a}}_{ij}) = 1$ we have:

$$h_{i,c,r}^{'0} = \sum_{j \in \mathcal{N}(i)} e^{-\frac{(||\mathbf{a}_{ij}||-r)^2}{2\delta}} \delta_{t_j,c} \tag{C2}$$

which correspond to $G_i^{(0)}(r,\delta,s)$ in [37]. Note that we require also no channel mixing at $l = 0$. For $l_F > 0$:

$$h_{i,c,r}^{'l_F} = \sum_{j \in \mathcal{N}(i)} \mathbf{Y}^{l_F}(\hat{\mathbf{a}}_{ij}) e^{-\frac{(||\mathbf{a}_{ij}||-r)^2}{2\delta}} \delta_{t_j,c} \tag{C3}$$

which, after a channel mixing step that sums over $c$ (mixing different particle types), correspond to $q_i^{(0)}(l,m,r,\delta)$ in [37]. By computing invariants from these features through their norm, we recover exactly $q_i^{(0)}(l,r,\delta)$ from [37].

By contrast, in our model we do not compute invariants after one layer, we keep equivariant features and let them interact over multiple layers in order to increase the expressivity.

Although this architecture qualitatively reproduces these expert features, for a quantitative match one would need to use a much larger cutoff radius $d_c = 5$ for building the graph, and a maximum rotational order of $l_{max} = 12$.

## Appendix D: Shiba vs Bapst dataset comparison

In Bapst's dataset [36], a timescale actually corresponds to a fixed value of the self-intermediate scattering function $F_k(t)$, so that different samples are measured at

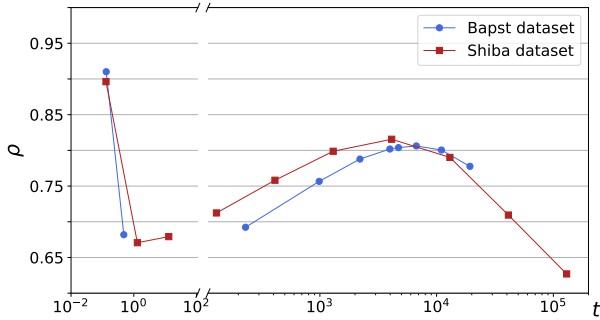

FIG. 15. **Comparison of datasets.** The same models trained at temperature $T = 0.44$ on each dataset.

slightly different times. Equilibration is performed under an NPT thermostat, *i.e.* at constant pressure and temperature, *i.e.* the volume is varying (and thus the density as well).

In Shiba's dataset [38], equilibration is performed at constant volume and temperature (NVT) so that the density is exactly $\rho = 1.2$ in all samples and at all temperatures. Furthermore, sampling of the trajectories is performed at fixed times, not fixed $F_k(t)$.

In Figure 15 we report the performance of our main model (thermal positions $+$ $E_{pot}(IS)$ inputs) for these two Kob-Anderson 3D datasets. Performances are shifted in time but otherwise rather comparable (a bit better on Shiba's dataset).