# Peer review of "Rotation-equivariant Graph Neural Networks for Learning Glassy Liquids Representations"

_SciPost Physics_

## Round 1 · Referee Report · Anonymous · 2023-12-5

Strengths

1- Proposal of a novel and innovative graph neural network (GNN) to account for rotational equivariance in glassy structures, which outperforms the state-of-the-art in learning structure-dynamics relationships.

2-Detailed and pedagogical introduction.

3-Extensive benchmarking which proofs the strong performance of the newly proposed GNN.

4-Clear discussion of the results and explanation of the strong performance of the method.

5-Extensive outlook for future work.

Weaknesses

1-Several times in the manuscript it is said “reduces the number of parameters”, “standard GNNs lack interpretability due [...] their high number of learnable parameters.”
I find this very confusing and potentially misleading, since these “standard” GNNs have 70k (Bapst) and 50k (BOTAN) parameters against the 50k parameters of the GNN presented in this manuscript. As a reader, I would expect at least a factor of 5-10 reduction in parameters from the way the manuscript is written. I would strongly suggest to remove all phrases which make statements along this line or disprove the numbers stated above.

2-Several times the GNNs of Bapst and Shiba are denoted as “standard”. This sounds a bit condescending in my opinion. I think the word “standard” can be dropped without changing the meaning of any of these sentences. If necessary just say “previous GNN approaches” or something along this line.

3-I was very troubled by the “discussion” in Section V (first two paragraphs on page 11 and 12), which contained, in my opinion, several very disputable statements which are also pretty much “off topic”. Examples:
“there is no doubt that a dynamical phase transition exists” (in my eyes there are very strong doubts).
“one observes a phase coexistence of mobile (liquid-like) parts and non mobile (solid-like) parts.” (I disagree)
“This coexistence phenomenon is commonly referred to as Dynamic Heterogeneities” (I disagree)
The authors need to be aware that “phase transition” and “phase coexistence” are very strong words and have very strict meaning in statistical physics. Dynamic heterogeneity just denotes that there are active and passive regions, but it does not need to imply “phase coexistence”.
In my eyes these paragraphs should be deleted since they are very questionable and also completely unnecessary for the manuscript.

4-Slightly along the lines of “C)” the authors use the word “structural order parameter” in a quite broad fashion. They explain this with the statement “Thus one obtains a formally purely structure-based function fθ({x}) that has a transition at the critical temperature, i.e. a structural order parameter.” But there is no critical temperature here (again this is a very clearly defined concept in statistical physics). We do not (yet) observe a phase transition between active and passive particles. My suggestion, again, would be to remove (most parts) of the first page of Section V and start with “Here we argue that a Deep Learning model fθ should be seen as a microscope…”. Additionally, I would replace the phrase “structural order parameter” by “learned structural descriptor” or something similar in the whole manuscript.

5-Some smaller points: 1) Typo on page 4 (“can be thought of”) 2) It would be nice to mark tau_alpha in Fig. 6. 3) I have seen that the Roadmap is on arXiv now, this could be updated.

Report

In the manuscript “Rotation-equivariant Graph Neural Networks for Learning Glassy Liquids Representations” Pezzicoli et al. propose a novel machine learning approach based on graph neural networks (GNNs) to systematically find structure-dynamics relationships in glassy liquids. The important upgrade proposed by the authors consists in using rotationally equivariant representations of the static structure. On the example of the Kob-Andersen mixture, which is a common benchmark in the field, the authors show that their new approach significantly outperforms existing GNNs and also other approaches based on MLPs or Ridge regression. The authors additionally investigate the usage of inherent states, transferability of trained networks to different temperatures and dynamic heterogeneities characterized by the dynamic susceptibility and the four-point correlation function.

The manuscript is (mostly) well written and also describes the theoretical foundations of the proposed equivariant GNNs in a pedagogical manner. The improvements compared to previous GNN designs are well described. The new approach is benchmarked against the state-of-the-art and performs very well. In particular, different time scales and the usage of inherent states are analyzed and compared to other approaches thus yielding a good understanding where the power of the newly developed GNN emerges from. Considering this assessment I generally recommend acceptance of this manuscript in “SciPost Physics” since the proposed rotation-equivariant GNNs could have an important impact on the field of machine learning glassy liquids. There are, however, several shortcomings which should be addressed before publication.

Requested changes

1-Correct statement on number of parameters

2-Remove phrase "standard"

3-Remove statements in Section V

4-Remove phrase "structural order parameter"

5-Some minor typos and corrections (see 5- above)

  • validity: top
  • significance: high
  • originality: top
  • clarity: high
  • formatting: perfect
  • grammar: good

Author:  François P. Landes  on 2024-03-13  [id 4363]

(in reply to Report 1 on 2023-12-05)

Sorry, the previous answer was badly formatted, I know found the workaround to make my markdown compliant.

Note: We have started to deposit the code and data on zenodo. The doi has been created but we have not yet made the repository public.

    ## Anonymous Report 1 on 2023-12-5 (Invited Report)

    ### Strengths

    1- Proposal of a novel and innovative graph neural network (GNN) to account for rotational equivariance in glassy structures, which outperforms the state-of-the-art in learning structure-dynamics relationships.

    2-Detailed and pedagogical introduction.

    3-Extensive benchmarking which proofs the strong performance of the newly proposed GNN.

    4-Clear discussion of the results and explanation of the strong performance of the method.

    5-Extensive outlook for future work.

First, we thank the referee for their quick and overall positive feedback. We dedicated time to attempt to provide a pedagogical introduction to rotation-equivariant layers and are pleased to see it was appreciated. Also we tried to provide clues as to why the network performs well (something that is too often missing in ML papers), so we are pleased to see it was appreciated.

We answer to the precise points below.

    ### Weaknesses

    1-Several times in the manuscript it is said “reduces the number of parameters”, “standard GNNs lack interpretability due [...] their high number of learnable parameters.”

I find this very confusing and potentially misleading, since these “standard” GNNs have 70k (Bapst) and 50k (BOTAN) parameters against the 50k parameters of the GNN presented in this manuscript. As a reader, I would expect at least a factor of 5-10 reduction in parameters from the way the manuscript is written. I would strongly suggest to remove all phrases which make statements along this line or disprove the numbers stated above.

Indeed, our discussion of this point was not clear. We changed the discussions about parameter numbers. Also there was an important typo in the appendix, that we corrected. As we now explain in the main text, because we use the same representation to predict 10 time scales and 2 particle types using the same backbone, our network is about an order of magnitude less parameter-hungry that other models, where each of the 10 time scales and each particle type need a dedicated network. Yet, this is not the main point and so we have soften the assertions about reducing the number of parameters.

    2-Several times the GNNs of Bapst and Shiba are denoted as “standard”. This sounds a bit condescending in my opinion. I think the word “standard” can be dropped without changing the meaning of any of these sentences. If necessary just say “previous GNN approaches” or something along this line.

Indeed, we meant "non rotation-equivariant" but "standard" could be thought as condescending. We removed all uses of this term.

    3-I was very troubled by the “discussion” in Section V (first two paragraphs on page 11 and 12), which contained, in my opinion, several very disputable statements which are also pretty much “off topic”. Examples:

“there is no doubt that a dynamical phase transition exists” (in my eyes there are very strong doubts). “one observes a phase coexistence of mobile (liquid-like) parts and non mobile (solid-like) parts.” (I disagree) “This coexistence phenomenon is commonly referred to as Dynamic Heterogeneities” (I disagree) The authors need to be aware that “phase transition” and “phase coexistence” are very strong words and have very strict meaning in statistical physics. Dynamic heterogeneity just denotes that there are active and passive regions, but it does not need to imply “phase coexistence”. In my eyes these paragraphs should be deleted since they are very questionable and also completely unnecessary for the manuscript.

Indeed, these matters are the subject of debate in the community, and the use of strong, precise words should be taken with caution. We have soften our assertions to reflect this.

Although this is not the main point of the paper, we would like to keep this discussion precisely because Machine Learning techniques challenge the usual way one thinks about these concepts. More precisely, we believe that the unprecedented ability of our network to generalize to unseen temperatures allows to open this discussion.

    4-Slightly along the lines of “C)” the authors use the word “structural order parameter” in a quite broad fashion. They explain this with the statement “Thus one obtains a formally purely structure-based function fθ({x}) that has a transition at the critical temperature, i.e. a structural order parameter.” But there is no critical temperature here (again this is a very clearly defined concept in statistical physics). We do not (yet) observe a phase transition between active and passive particles. My suggestion, again, would be to remove (most parts) of the first page of Section V and start with “Here we argue that a Deep Learning model fθ should be seen as a microscope…”. Additionally, I would replace the phrase “structural order parameter” by “learned structural descriptor” or something similar in the whole manuscript.

As said above, we would like to keep this discussion, although it's partly speculative. In the resubmission, we keep the wording “structural order parameter” but we checked carefully that each time we use it, it's in the conditional form, or some other form softer than an assertive statement.

    5-Some smaller points: 1) Typo on page 4 (“can be thought of”) 2) It would be nice to mark tau_alpha in Fig. 6. 3) I have seen that the Roadmap is on arXiv now, this could be updated.

Thank you for spotting out these, they are now corrected

Author:  François P. Landes  on 2024-03-13  [id 4361]

(in reply to Report 1 on 2023-12-05)
Category:
answer to question
reply to objection
correction

Referee reports and our answers

System Message: WARNING/2 (<string>, line 2)

Title underline too short.

Referee reports and our answers
==============================

Note: We have started to deposit the code and data on Zenodo. The DOI has been created but we have not yet made the repository public.

---

## Round 1 · Referee Report · Anonymous · 2023-12-5

Strengths

  1. Proposal of a novel and innovative graph neural network (GNN) to account for rotational equivariance in glassy structures, which outperforms the state-of-the-art in learning structure-dynamics relationships.

  2. Detailed and pedagogical introduction.

  3. Extensive benchmarking which proofs the strong performance of the newly proposed GNN.

  4. Clear discussion of the results and explanation of the strong performance of the method.

  5. Extensive outlook for future work.

    First, we thank the referee for their quick and overall positive feedback. We dedicated time to attempt to provide a pedagogical introduction to rotation-equivariant layers and are pleased to see it was appreciated. Also we tried to provide clues as to why the network performs well (something that is too often missing in ML papers), so we are pleased to see it was appreciated.

    We answer to the precise points below.

Weaknesses

  1. Several times in the manuscript it is said "reduces the number of parameters", "standard GNNs lack interpretability due [...] their high number of learnable parameters." I find this very confusing and potentially misleading since these "standard" GNNs have 70k (Bapst) and 50k (BOTAN) parameters against the 50k parameters of the GNN presented in this manuscript. As a reader, I would expect at least a factor of 5-10 reduction in parameters from the way the manuscript is written. I would strongly suggest to remove all phrases which make statements along this line or disprove the numbers stated above.

    Indeed, our discussion of this point was not clear. We changed the discussions about parameter numbers. Also there was an important typo in the appendix that we corrected. As we now explain in the main text, because we use the same representation to predict 10 time scales and 2 particle types using the same backbone, our network is about an order of magnitude less parameter-hungry that other models, where each of the 10 time scales and each particle type need a dedicated network. Yet, this is not the main point and so we have softened the assertions about reducing the number of parameters.

  2. Several times the GNNs of Bapst and Shiba are denoted as "standard". This sounds a bit condescending in my opinion. I think the word "standard" can be dropped without changing the meaning of any of these sentences. If necessary, just say "previous GNN approaches" or something along this line.

    Indeed, we meant "non rotation-equivariant" but "standard" could be thought of as condescending. We removed all uses of this term.

  3. I was very troubled by the "discussion" in Section V (first two paragraphs on page 11 and 12), which contained, in my opinion, several very disputable statements which are also pretty much "off-topic". Examples: "there is no doubt that a dynamical phase transition exists" (in my eyes, there are very strong doubts). "one observes a phase coexistence of mobile (liquid-like) parts and non-mobile (solid-like) parts." (I disagree) "This coexistence phenomenon is commonly referred to as Dynamic Heterogeneities" (I disagree). The authors need to be aware that "phase transition" and "phase coexistence" are very strong words and have a very strict meaning in statistical physics. Dynamic heterogeneity just denotes that there are active and passive regions, but it does not need to imply "phase coexistence". In my eyes, these paragraphs should be deleted since they are very questionable and also completely unnecessary for the manuscript.

    Indeed, these matters are the subject of debate in the community, and the use of strong, precise words should be taken with caution. We have softened our assertions to reflect this.

    Although this is not the main point of the paper, we would like to keep this discussion precisely because Machine Learning techniques challenge the usual way one thinks about these concepts. More precisely, we believe that the unprecedented ability of our network to generalize to unseen temperatures allows us to open this discussion.

  4. Slightly along the lines of "C)", the authors use the word "structural order parameter" in a quite broad fashion. They explain this with the statement "Thus one obtains a formally purely structure-based function fθ({x}) that has a transition at the critical temperature, i.e. a structural order parameter." But there is no critical temperature here (again this is a very clearly defined concept in statistical physics). We do not (yet) observe a phase transition between active and passive particles. My suggestion, again, would be to remove (most parts) of the first page of Section V and start with "Here we argue that a Deep Learning model fθ should be seen as a microscope…". Additionally, I would replace the phrase "structural order parameter" by "learned structural descriptor" or something similar in the whole manuscript.

    As said above, we would like to keep this discussion, although it's partly speculative. In the resubmission, we keep the wording "structural order parameter" but we have checked carefully that each time we use it, it's in the conditional form, or some other form softer than an assertive statement.

  5. Some smaller points: 1) Typo on page 4 (“can be thought of”) 2) It would be nice to mark tau_alpha in Fig. 6. 3) I have seen that the Roadmap is on arXiv now, this could be updated.

    Thank you for spotting these, they are now corrected.

---

## Round 1 · Referee Report · Joerg Rottler · 2023-12-19

Strengths

1. advances ML techniques for studying glassy dynamics with a challenging implementation of graph neural networks
2. uses published datasets so direct comparison with earlier work is possible
3. explores transferability of the fitted models

Weaknesses

1. quite technical, likely accessible only for expert audience that have followed much of the previous work in the glassy ML domain
2. differences or advances between proposed implementation and other codes not entirely clear
3. figures poorly formatted

Report

The manuscript develops a neural network model to predict the dynamic propensity of particles in supercooled glassy liquids from structure via supervised machine learning. It extends previous work based on graph neural network by adding the notion of rotational equivariance. This additional symmetry appears to improve both performance and transferability across different temperatures, thus establishing a new "state-of-the-art".
The work is timely and a logical next step in the current efforts to leverage machine learning methods in glassy physics. The results are important and should be communicated. Below I offer (mostly optional) suggestions for further improvement of the manuscript, and for making it more accessible to a broader audience. Currently the paper is likely accessible to an expert audience only.
1) Since the main development is the introduction of rotational equivariance in the GNNs, the physical significance of this concept could be better explained and highlighted more in the introduction. Why is it needed and how does it interact with rotational invariance? The paragraph on the early SVMs and softness could instead be omitted as it feels more like a review.
2) Sections II and III are very technical and challenging to appreciate for readers unfamiliar with computer vision and SE(3) invariant techniques. First, is there anything "specifically glassy" in this development? Fig. 2 is helpful but generic, could an example be given from one of the actual simulations? Any additional intuition would also be helpful. In general, it is difficult to appreciate which parts of the development are original to this work, and which parts are generic for any descriptions of molecules and condensed matter systems, and as such have essentially appeared already in refs [42-49] in some form. It seems to me that spherical harmonics, CG tensor producs, conv. layers are all standard things, should those go in an appendix?
Why do we need a new equivariant architecture, in what sense is it specialized? Couldn't we just take or adapt the code of ref [46] for instance (developed for fitting interatomic potentials) and regress against propensity instead of local forces/energies? Is there anything fundamentally different between the authors' implementation and these codes?
3) Fig. 6 shows that the proposed GNN performs better than all previous methods. Is it possible to give more insight into why that is the case? What specific new physics is responsible (beyond just saying "one more symmetry respected")?
The comparison is a bit problematic. Although nominally the same correlation is plotted, there is variation not only in the ML method but also in the underlying dataset. In particular, correlations for the two NVT datasets deteriorate at large times, why is this so? I note that the much simpler linear regression on expert features by Alkemade (2023) is better than the authors' method for all times smaller than 2000. Could the authors use this linear regression method on their NPT dataset and show us the results? Perhaps this simple linear model is already "almost as good" as the much more complicated equivariant GNN model. If one does not want to squeeze out a few more percentage points of correlation, then one might prefer the LR model which is probably much faster to fit and also fully interpretable. I would find this an important insight to give.
4) Figs. 5-15 are of very poor quality. The fonts are so small that it is almost impossible to read the axis labels and legends. Please increase the font size substantially on all plots. Fig. 9 is particularly bad, and I am honestly not sure what conclusions I am supposed to draw from it. Symbols overlap and there seem to be more symbols than given in the legend. Please rework this figure completely.
5) There are very successful physical order parameters that correlate with local mobility upon some coarse-graining, for instance ref.[1], the 'theta-parameter'. Could the authors compute this for their dataset and add to Fig. 6? It would seem a fair challenge to ML based approaches.
6) What do the authors say about the recent contribution by Oyama et al. Frontiers in Physics 10:1007861, doi:10.3389/fphy.2022.1007861? They seem to achieve similar correlation coefficients of ~0.8 with an augmented CNN instead. This begs the question about what is really the important, minimal physics that is needed to anticipate particle mobility from local structure? Is there an upper ceiling to the correlation coefficent, is there some systematic way to say that now we have converged?

Requested changes

see report

  • validity: top
  • significance: high
  • originality: high
  • clarity: good
  • formatting: good
  • grammar: excellent

Author:  François P. Landes  on 2024-03-13  [id 4362]

(in reply to Report 2 by Joerg Rottler on 2023-12-19)

---

## Round 1 · Referee Report · Joerg Rottler · 2023-12-19

### Strengths

    1. advances ML techniques for studying glassy dynamics with a challenging implementation of graph neural networks
    2. uses published datasets so direct comparison with earlier work is possible
    3. explores transferability of the fitted models

    ### Weaknesses

    1. quite technical, likely accessible only for expert audience that have followed much of the previous work in the glassy ML domain
    2. differences or advances between proposed implementation and other codes not entirely clear
    3. figures poorly formatted
* * *
    ### Report

    The manuscript develops a neural network model to predict the dynamic propensity of particles in supercooled glassy liquids from structure via supervised machine learning. It extends previous work based on graph neural network by adding the notion of rotational equivariance. This additional symmetry appears to improve both performance and transferability across different temperatures, thus establishing a new "state-of-the-art".
    The work is timely and a logical next step in the current efforts to leverage machine learning methods in glassy physics. The results are important and should be communicated. Below I offer (mostly optional) suggestions for further improvement of the manuscript, and for making it more accessible to a broader audience. Currently the paper is likely accessible to an expert audience only.

We thank the referee for their overall positive feedback.

    1) Since the main development is the introduction of rotational equivariance in the GNNs, the physical significance of this concept could be better explained and highlighted more in the introduction. Why is it needed and how does it interact with rotational invariance? The paragraph on the early SVMs and softness could instead be omitted as it feels more like a review.

Indeed, our introduction starts with a history of ML for glasses, and was not explaining what we mean by rotation equivariance. We have now inserted the following paragraph in the introduction: "Concretely, under rotation of the whole glass, the scalar properties of the particles (as mobility) remained unchanged, while the vectorial quantities (like relative positions) transform accordingly. With SE(3)-equivariant networks, the internal representations behave like such physical vectors: the representation rotates appropriately under rotation of the input. Invariant features have significantly less expressivity than equivariant ones, since they are just a subset of those."

    2) Sections II and III are very technical and challenging to appreciate for readers unfamiliar with computer vision and SE(3) invariant techniques.

We are sorry to hear that. We aimed at providing a pedagogical introduction to the field, introducing only the necessary concepts to understand our network, assuming some background in Deep learning. For SE(3) techniques, we tried to provide a minimal but self-sufficient introductory presentation about these, so that the reader could appreciate the novelty of our proposed GNN. We note that he other referee's feedback was more positive on the pedagogical aspect of our work. We would be happy to clarify if some precise points are unclear, keeping in mind that the paper does not intend to be a self-sufficient deep learning introductory course.

    Fig. 2 is helpful but generic, could an example be given from one of the actual simulations? Any additional intuition would also be helpful.

As our internal representation (even at the first layer) is a set of 8 feature vectors (each channel can be thought of as a 3D vector), we could plot one of them, but this would result in a messy figure. Here the point is to remember that vectors transform non-trivially under rotation (the point to which they are attached moves, but the vector itself has to rotate too).

    First, is there anything "specifically glassy" in this development? 
    (...)
    In general, it is difficult to appreciate which parts of the development are original to this work, and which parts are generic for any descriptions of molecules and condensed matter systems, and as such have essentially appeared already in refs [42-49] in some form. 
    (...)
    Why do we need a new equivariant architecture, in what sense is it specialized? Couldn't we just take or adapt the code of ref [46] for instance (developed for fitting interatomic potentials) and regress against propensity instead of local forces/energies? Is there anything fundamentally different between the authors' implementation and these codes?

    First, is there anything "specifically glassy" in this development?

To answer very crudely: no, not really. Then, the details of implementation are specific to our task, and can make a difference (encoding of the particle types, choice of radial basis encoding, skip connections of the input to each layer, etc). Furthermore, there are some original choices in our work, such as using a shared backbone (representation) for all time scales and particle types, or using the potential energy as node feature. These choices shape the task and the global architecure, and were indeed inspired by our understanding of glassy physics.

To be more clear about our contributions, in sec II (and III.B,D) we review known facts about equivariant networks, and in sec III (A,C) and section IV we present our choice of how to combine them. As often in ML works, each network is unique although some of the choices may not be crucial. In section IV we introduce the truly original variations that are specific to our architecture (as the multi-variate regression or inclusion of potential energy as node feature).

    Couldn't we just take or adapt the code of ref [46] for instance

In essence, yes, altough the adaptations presented in sec III and IV would be needed. We do suggest this would be a good test, as [46] introduces a very powerful kind of equivariant layer.

    It seems to me that spherical harmonics, CG tensor producs, conv. layers are all standard things, should those go in an appendix?

We decided to include those in the main text so as to provide a self-contained introduction to SE(3)-equivariant GNNs, so that our method can be understood. One cannot understand how equivariant is more expressive than invariant without going through this introduction, and so we deem this is an important part of the main text.

    3) Fig. 6 shows that the proposed GNN performs better than all previous methods. Is it possible to give more insight into why that is the case? What specific new physics is responsible (beyond just saying "one more symmetry respected")?

Our main point is indeed that the crucial point is to respect one more symmetry (a non trivial one in practice). Some of the previous methods do use rotation-invariant descriptors (such as the Bond Order parameter [see Coslovich's works] or Laura Filion's group descriptors). However, here we are able to combine neighboring particles' equivariant descriptors to build more detailed representations. Currently, it is difficult for us to explain any better than this. Understanding better how these network are able to perform well is road for future work.

Then, about what specific new physics is responsible [for performance], we show that using the information about inherent structures (IS) is helpful too, as we explain in sec. IV.C. And we show that including potential energy is a trick to include information about the IS at a small cost (and, by itself, slightly improves performance) -- this is also discussed in sec. IV.C. We discuss a bit the intuitive explanations for these results in that same section.

    The comparison is a bit problematic. Although nominally the same correlation is plotted, there is variation not only in the ML method but also in the underlying dataset.

Indeed, we agree that a fair comparison can only be performed using the same data (same system, temperatures, equilibration protocols, etc). Previous works sometimes compared different glass-forming models and we wanted to outline that this is potentially misleading (see sec. IV.D): Often, comparison with previous works can not be done rigorously, for two reasons: use of different datasets and different input data. As mentioned in the previous section and already pointed out in [40] the two main datasets [36, 38] differ in many features (see Appendix D for details), although being built from MD simulations of the same system (3D Kob-Andersen mixture). and later: To compare Machine Learning architectures in a fair way, one should work at fixed task (fixed dataset and in- put data). We now respect this constraint to obtain two precise results, that we deem significant.

    In particular, correlations for the two NVT datasets deteriorate at large times, why is this so?

Indeed, for both datasets, either equilibrated under NVT or NPT thermostat, prediction accuracy degrades at longer times. Although the (iso-configurational average) propensity is well-defined even at long times, the way in which future dynamics depends on t=0 structure probably becomes more and more non-local, so that even a relatively deep network (7 layers) cannot capture the proper correlations.

Why the NPT-equilibrated data seems to be more predictable (slower decay of \rho) than the NVT-prepared, is not clear to us. As we mention in the text, there are multiple differences between the 2 datasets, so the culprit may not even be the thermostat chosen. We do not know.

    I note that the much simpler linear regression on expert features by Alkemade (2023) is better than the authors' method for all times smaller than 2000. Could the authors use this linear regression method on their NPT dataset and show us the results? Perhaps this simple linear model is already "almost as good" as the much more complicated equivariant GNN model. If one does not want to squeeze out a few more percentage points of correlation, then one might prefer the LR model which is probably much faster to fit and also fully interpretable. I would find this an important insight to give.

Such a rigorous comparison (using same dataset) has now been performed in the Roadmap (ref [57]), on the NVT dataset. The results are qualitatively the same: Alkemade (2023) (ref [37] for us) indeed outperforms our method at all times smaller than 2000. However, another very important difference lies in the input features used in Alkemade (2023). Indeed, that work uses the size of the particle' local cage as input feature, something which is rather costly/complicated to define/compute, and essentially contains the target label for short times.

As stated in the Roadmap [57], CAGE [Alkemade (2023)] performs best for shorter times, which appears reasonable as it uses extensive Monte- Carlo simulations to characterize the local cage struc- ture at short times.

Admitedly, we tried to mimick this feature but failed to see any improvement. As we mention quickly in the text: using as input node feature the distance to the local cage center (estimated as the quenched position),(...) our attempts did not yield any significant improvement (nor deteriorated the performance). Our failure may be due to implementation details, or to the fact we used the IS position as proxy for the cage center, and the difference between current and IS position as proxy for the cage size, something that is not as good as actually simulating the cage (this was discussed in [Alkemade (2023)]).

Note that we study the connection between our approach and Ref [37]'s approach in appendix C. In page 6, we refer to appendix C: This projection can be seen as an embedding. Taking the norm of this representation, one would qualitatively obtain the expert descriptors described in [37] (for an exact and more detailed derivation, see Appendix C).

    4) Figs. 5-15 are of very poor quality. The fonts are so small that it is almost impossible to read the axis labels and legends. Please increase the font size substantially on all plots.

Indeed, we overlooked to check this matter. We have now increased font size for axis labels and legends.

    Fig. 9 is particularly bad, and I am honestly not sure what conclusions I am supposed to draw from it. Symbols overlap and there seem to be more symbols than given in the legend. Please rework this figure completely.

Indeed, Fig 9 was confusing at first sight. We have now clarified in the caption the interpretation of the legend by adding: "For each training temperature (color) two different experiments are performed: transfer test (square markers with dashed lines) and transfer learn (upper triangle with full line). This results in 8 curves per plot coming from all the combinations of colors and line-styles."

We cropped the content to show only the relevant values of \rho (hiding the very small values) to help better read the markers. Symbols can still overlap due to the choice of the scale, which is fixed in order to allow fair comparison.

    5) There are very successful physical order parameters that correlate with local mobility upon some coarse-graining, for instance ref.[1], the 'theta-parameter'. Could the authors compute this for their dataset and add to Fig. 6? It would seem a fair challenge to ML based approaches.

Indeed, this is a fair challenge. However, the 'theta-parameter' \Theta of Tong and Tanaka does not easily transfer to 3D systems. A rigorous comparison was done in 2D in the Roamap [57], where \Theta is actually substantially worse than supervised methods, as can be seen in Fig 3 of that paper (or Fig 4, which report the Bond-Breaking propensity). In page 23 of [57], the difference between correlations reported in previous papers and the one reported in [57] is partly explained: In this roadmap, we show that the per- formance is significant reduced when applied to systems which are not prone to crystallization. Additionally, some of the performance difference might emerge from calculating Pearson correlation over all particles indepen- dent of their radii in Ref. [S16], as opposed to making independent predictions for each particle type as done in this manuscript.

    6) What do the authors say about the recent contribution by Oyama et al. Frontiers in Physics 10:1007861, doi:10.3389/fphy.2022.1007861? They seem to achieve similar correlation coefficients of ~0.8 with an augmented CNN instead.

Thank you for pointing out this interesting reference. In this paper however, there are multiple differences with our task. Their model (KAM) is 2D and not 3D, the temperatures are different, and quite importantly, the propensity is coarse-grained over a length scale. To compare fairly, one would need to compare the correlations computed on the same labels, i.e. the propensity (not spatially coarse-grained), on the same model (3D KA, at the same temperatures).

Furthermore, the fact that the top correlation with propensity of their CNN+Grad-CAM model (\Gamma) is almost as good as the one obtained from simply computing the Voronoi volume \Upsilon (fig S8 in their Supp. Mat.) calls for caution, as in our setup (which has been used in many papers), the free volume is a bad predictor of the dynamics.

However, it is very interesting to see that coarse-grained mobility is somewhat intrinsically easier to predict than single-particle mobility, and we now mention this as interesting direction for future work in our revised manuscript: sec VI: ''use coarse-grained mobility measures as target labels. Indeed, they have been shown to be more structure-dependent [50] and thus also display larger correlation coefficients with simple structural descriptors [67]. Eliminating noise in the labels could help achieve better precision, possibly reaching the maximum''

    This begs the question about what is really the important, minimal physics that is needed to anticipate particle mobility from local structure?

We want to stress here that the model they propose is as black-box as ours, while other (interpretable) descriptors like the Tong-Tanaka Order parameter \Theta or Voronoi volume \Upsilon are shown to perform much worse than ours under fair comparison (see our ref [57], Roadmap on machine learning glassy liquids). Our approach aims to bridge the gap between expert features and deep models (thus going into the direction of more interpretable models): see section II.G of our manuscript.

    Is there an upper ceiling to the correlation coefficent, is there some systematic way to say that now we have converged?

There is an upper ceiling, but this limit only comes from the convergence (or not) of the iso-configurational average. Using a finite number of copies of the system to average over, one obtains an estimation for the true average (with inifinitely many copies). The deviation from this converged value can be estimated using bootstrap, and is reported in the Roadmap (ref [57]) as shaded area in the top part of the correlation plots.

---

## Round 2 · Referee Report · Anonymous (Referee 1) · 2024-3-21

Report

The authors have answered in detail to the critique and have revised their manuscript accordingly.

Since the method proposed in the manuscript opens a new pathway in learning structure-dynamics correlations in supercooled liquids with interesting applications for better understanding dynamic heterogeneity, I recommend publication of "Rotation-equivariant Graph Neural Networks for Learning Glassy Liquids Representations" in SciPost Physics.

---

## Round 2 · Referee Report · Joerg Rottler (Referee 2) · 2024-3-28

Report

I appreciate the authors' thoughtful responses to my first report. The manuscript has also been improved in multiple places and can now be accepted. It would have been nice to see some of the authors' responses added to the main text so that other readers also benefit. Right now, only absolutely required modifications were made. They could still take up that opportunity when they submit their final version should they so decide.

---

## Round 2 · Author Response

Note: We have started to deposit the code and data on zenodo. The doi has been created but we have not yet made the repository public.

We have replied to the referees.

---

## Round 2 · List of Changes

We upload a pdf with changes highlighted in blue for convenience.

---

## Editorial Decision

resubmitted